# Predictive Prefetching for Retrieval-Augmented Generation

**Wuyang Zhang** [1]   **Shichao Pei** [1]

## Abstract

Retrieval-Augmented Generation (RAG) improves factual grounding in large language models but suffers from substantial latency due to synchronous retrieval. While recent work explores asynchronous retrieval, existing approaches rely on heuristic coordination between retrieval and generation and assume stable information demands during decoding that often break in complex, multi-domain settings. In this paper, we propose an advanced asynchronous retrieval framework that enables predictive prefetching aligned with evolving information needs. The framework explicitly predicts when retrieval should be triggered and what information should be retrieved using three components, a retrieval predictor, a context monitor, and a query generator, by exploiting semantic precursors in generation dynamics that emerge several tokens before uncertainty becomes critical. Experiments on multiple benchmarks demonstrate up to 43.5% end-to-end latency reduction and 62.4% improvement in time-to-first-token, while maintaining answer quality comparable to synchronous RAG baselines.

## 1. Introduction

Retrieval-Augmented Generation (RAG) has become the dominant approach for grounding large language model (LLM) outputs in factual and up-to-date information (Lewis et al., 2020; Guu et al., 2020; Borgeaud et al., 2022). By augmenting static model parameters with external knowledge, RAG effectively mitigates hallucination and knowledge staleness. However, the latency overhead introduced by retrieval operations remains a major barrier to production deployment and significantly degrades user experience. Complex queries often require multiple sequential retrieval steps (Asai et al., 2024; Jiang et al., 2023), particularly

deep research workloads (Yao et al., 2023; Trivedi et al., 2023) may trigger hundreds of retrievals within a single query. Moreover, in high-stakes applications such as financial analysis, news summarization, and emergency response, retrieval frequently relies on external data sources rather than pre-indexed local collections. These external calls typically incur 100–500 ms latency per retrieval due to network round trips and API rate limits, causing the cumulative overhead to become prohibitive.

The core architectural limitation stems from the *synchronous* nature of current RAG designs. When uncertainty triggers a retrieval, token generation is fully suspended until the retrieval completes (Kwon et al., 2023). This reactive behavior creates a fundamental conflict between generation quality and system performance. Applications that demand high factual accuracy must tolerate multiple retrieval rounds and cumulative delays, whereas latency-sensitive deployments are forced to limit retrieval depth, inevitably sacrificing the completeness and reliability of retrieved evidence.

Despite the growing number of advanced retrieval mechanisms, little attention has been devoted to addressing this fundamental architectural limitation. Recent concurrent work has begun to explore asynchronous retrieval strategies (Lin et al., 2025; Wang et al., 2025; Jiang et al., 2025). However, these approaches largely depend on manually designed heuristics to coordinate retrieval and generation, such as triggering retrieval at fixed time intervals or using stale tokens as queries, assuming relatively stable information demands during generation. While this assumption holds in scenarios with high query–document similarity, it becomes fragile in complex multi-domain settings, where topic boundaries blur, and entity references evolve midgeneration. As a result, prefetching may introduce irrelevant context, undermining both generation efficiency and factual reliability. In such cases, current asynchronous architectures primarily mask retrieval latency but cannot correct mismatches between retrieved content and actual information needs, potentially leading to increased computation and delayed responses.

Motivated by the limitation of existing asynchronous retrieval strategies, in this paper, we propose an advanced asynchronous retrieval framework that explicitly addresses three key questions: ***when retrieval should be triggered***

---

[1]Department of Computer Science, University of Massachusetts Boston, Boston, United States of America. Correspondence to: Shichao Pei <shichao.pei@umb.edu>.

*Proceedings of the 43rd International Conference on Machine Learning*, Seoul, South Korea. PMLR 306, 2026. Copyright 2026 by the author(s).

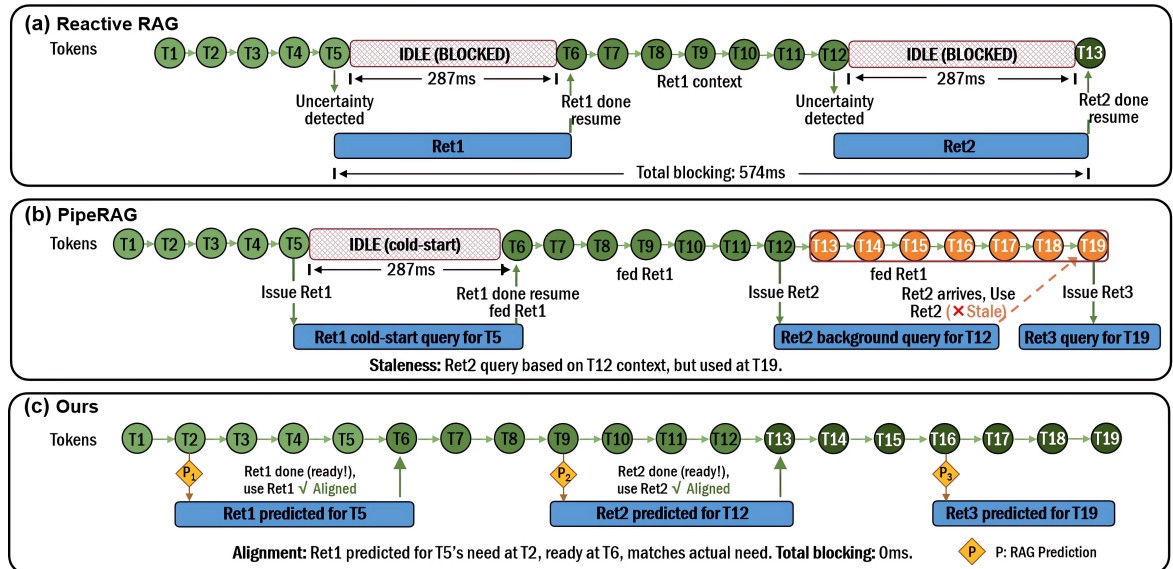

*Figure 1.* **Comparison of RAG architectures.** (a) *Synchronous RAG*: generation blocks during each retrieval (287ms). (b) *PipeRAG*: retrieval runs in parallel but suffers from query staleness, since queries use outdated context (e.g., RET$_2$ at token 12 is applied at token 16 when context has shifted). (c) *Ours*: predictive prefetching anticipates future needs. At token 2, we predict token 5 will require retrieval and issue a query for token 5's anticipated context. Retrieved content arrives aligned with actual needs, reducing end-to-end latency by 43.5%.

*to initiate prefetching,* **whether** *the current context is sufficient to support retrieval, and* **what** *information should be retrieved to assist generation.* Our key insight is twofold. First, retrieval needs are preceded by identifiable semantic precursors in generation dynamics, such as characteristic patterns in entropy trajectories, attention allocation, and value representation dynamics, which emerge approximately 8–16 tokens before uncertainty becomes critical. Second, these same signals encode retrieval intent and can be leveraged to infer the retrieval query itself.

Building on these insights, to determine retrieval timing, we introduce a *retrieval predictor* that forecasts impending information needs by monitoring generation signals, including token distributions, attention patterns, and discourse markers. Then, to assess whether the accumulated generation context provides adequate semantic information for reliable query construction, we design a *context monitor* to determine the optimal waiting horizon before initiating retrieval. Lastly, a *query generator* is developed to construct queries aligned with anticipated information requirements rather than merely echoing recent context.

Together, these components enable informed decisions about both when to initiate retrieval and what information to retrieve. This predictive capability supports asynchronous prefetching, allowing retrieval to proceed concurrently while generation continues uninterrupted, such that the prefetched content aligns with the model's actual information needs when uncertainty arises. Specifically, all three components

(the retrieval predictor, context monitor, and query generator) are pretrained jointly on collected historical generation traces and further adapted online via policy gradient with action-specific feedback. Through this process, the predictor learns to make prefetching decisions that improve generation quality, balancing prediction confidence with resource constraints.

Our main contributions are summarized as follows:

- We propose an asynchronous retrieval strategy that explicitly predicts when retrieval should be triggered and what information should be retrieved to support generation.
- We design three key components, namely a retrieval predictor, a context monitor, and a query generator, that enable asynchronous prefetching and align retrieved content with the model's actual information needs.
- A comprehensive evaluation on HotpotQA, 2WikiMultiHopQA, Natural Questions, and TriviaQA, demonstrates 43.5% end-to-end latency reduction, 62.4% time-to-first-token improvement, and 31% fewer retrievals per 1K tokens, while maintaining answer quality within 1% of synchronous RAG baselines.

## 2. Related Work

Retrieval-Augmented Generation (RAG) enhances language models by incorporating external knowledge, but each retrieval introduces additional latency that directly constrains generation throughput. Prior work has addressed this bottle-

*Table 1.* Positioning of adaptive retrieval methods. Our approach uniquely combines learned predictive timing with asynchronous execution and semantic query optimization.

| Method | Timing | Execution | Decision | Query Signal |
|---|---|---|---|---|
| Self-RAG | Reactive | Sync | Learned | Raw tokens |
| FLARE | Reactive | Sync | Heuristic | Raw tokens |
| DRAGIN | Reactive | Sync | Heuristic | Attention-weighted |
| Adaptive-RAG | Reactive | Sync | Learned | Raw tokens |
| TeleRAG | Fixed Interval | *Async* | Heuristic | Raw tokens |
| PipeRAG | Reactive | *Async* | Heuristic | Raw tokens |
| **Ours** | **Predictive** | ***Async*** | **Learned** | **Semantic-based** |

neck along two main directions: system-level optimization and adaptive retrieval strategy. We situate our predictive prefetching approach with respect to both lines, with detailed technical discussion provided in Appendix H.

**System Optimization.** System-level optimizations reduce retrieval overhead through advances such as efficient vector databases (Johnson et al., 2021), hybrid retrieval that combines dense and sparse signals, knowledge graph–based methods (Edge et al., 2025), and cache-augmented strategies (Chan et al., 2025). Despite shortening per-query latencies, these approaches still maintain a dependency between retrieval and generation, leaving generation fundamentally constrained by retrieval latency. Our predictive prefetching approach is orthogonal to such optimizations: by forecasting retrieval needs in advance, it overlaps retrieval with ongoing generation, effectively hiding retrieval latency.

**Adaptive Retrieval Strategy.** Adaptive methods such as FLARE (Jiang et al., 2023), Self-RAG (Asai et al., 2024), and DRAGIN (Su et al., 2024) dynamically trigger retrieval based on uncertainty signals, but do so reactively, first detecting uncertainty and then blocking generation to perform retrieval. TeleRAG (Lin et al., 2025) and PipeRAG (Jiang et al., 2025) are concurrent efforts that explore asynchronous retrieval via pipeline parallelism. TeleRAG relies on a fixed lookahead window, triggering retrieval at predetermined intervals even when additional retrieval is unnecessary. PipeRAG instead uses generated and potentially stale context as retrieval queries, which can misalign retrieved information with the model's evolving semantic needs. None of these methods predict *what* to retrieve based on the evolving semantics during generation, resulting in topic mismatch, hallucination from irrelevant context, and longer generation due to self-correction, ultimately increasing latency, computational cost, and degrading response quality. Table 1 provides a detailed comparison between these approaches.

## 3. Preliminaries

**Design Principles.** Our framework is guided by three core principles:

- *Temporal Decoupling.* Unlike traditional RAG systems, where retrieval synchronously blocks generation, we should decouple retrieval from generation and execute the two processes in parallel. A retrieval predictor anticipates retrieval needs 8–16 tokens ahead, allowing retrieval to complete asynchronously while generation proceeds uninterrupted.

- *Minimal Overhead.* The framework should introduce negligible computational overhead. It operates as a lightweight auxiliary module and reuses existing generation signals, ensuring that overall system throughput remains unaffected.

- *Fallback and Caching.* To ensure robustness, the system should degrade to synchronous retrieval when predictions fail, or retrieval arrives late, while unused prefetched documents are retained to amortize retrieval costs across subsequent generation steps.

**Design Assumptions.** Our approach assumes access to LLM hidden states and attention weights at inference time and does not require LLM retraining or fine-tuning. We target retrieval latencies in the range of 100–500 ms, typical of external APIs and vector databases, where latency hiding yields substantial benefit. Uncertainty events are defined by entropy threshold crossings.

## 4. Methodology

Our framework fundamentally decouples retrieval from generation through an asynchronous architecture that enables concurrent execution of both processes. Figure 2 illustrates the core components, the retrieval predictor, context monitor, and query generator, and their interactions within the system. We pretrain these components jointly using a multi-task learning objective, and subsequently adopt online policy-gradient adaptation with action-specific feedback to continuously adjust retrieval behaviors after deployment.

### 4.1. Retrieval Predictor

○ **When should retrieval be triggered to prefetch?**

Decoupling retrieval from generation requires determining *when* to initiate retrieval before uncertainty necessitates additional context. We therefore develop a retrieval predictor that predicts retrieval needs within a lookahead horizon by leveraging internal LLM representations, which capture evolving semantic abstractions, attention patterns, and uncertainty signals critical for anticipating future information needs beyond what token-level heuristics can provide.

Specifically, the retrieval predictor estimates the probability that retrieval will be needed within a lookahead horizon of

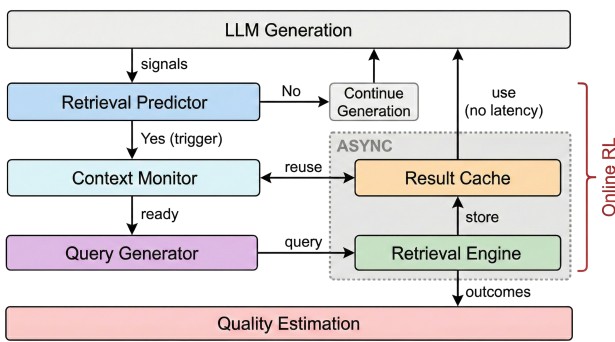

*Figure 2.* The architecture decoupling generation and retrieval. Retrieval Predictor monitors transformer signals, Context Monitor assesses query readiness, and Query Generator initiates asynchronous retrieval. Documents and embeddings are stored in a shared Result Cache. Online learning (red) adapts all three components based on retrieval outcomes.

$\Delta$ tokens:

$$\hat{p}_t = \textbf{RetrievalPredictor}(\mathbf{H}_t, \mathbf{A}_t, \mathbf{V}_t, \mathbf{o}_t) \in [0, 1] \tag{1}$$

where $\hat{p}_t$ estimates the likelihood that token-level entropy $\mathcal{H}_\tau$ will first exceed threshold $\theta$ at some position $\tau \in [t+1, t+\Delta]$. where $\mathbf{H}_t$, $\mathbf{A}_t$, and $\mathbf{V}_t$ denote internal LLM representations from middle-upper layers (i.e., 30–45% of model depth), and $\mathbf{o}_t$ contains output distribution statistics (entropy, top-k probability margins). In particular, $\mathbf{H}_t$ represents hidden states over a 16-token sliding window, $\mathbf{A}_t$ denotes attention weight matrices capturing focus patterns, and $\mathbf{V}_t$ captures value vectors from attention for tracking information flow changes. We select this depth range because interpretability studies (Rogers et al., 2020) show that intermediate layers capture high-level semantic abstractions while preserving uncertainty-related signals, whereas final layers tend to overfit to output distributions. For example, layers 10–14 in a 32-layer Llama-3.1-8B.

The entropy threshold $\theta$ determines what constitutes high uncertainty warranting retrieval. Higher thresholds reduce retrieval frequency but may miss beneficial opportunities; lower thresholds increase coverage at the cost of more triggers. The threshold can be tuned per application: latency-sensitive deployments prefer higher $\theta$, quality-critical applications prefer lower values. Specific threshold selection and sensitivity analysis appear in Appendix E.6.

**RetrievalPredictor Architecture.** We employ a lightweight 2-layer transformer encoder for uncertainty prediction. The encoder processes the concatenated signal representations as follows:

$$\mathbf{z}_t = \text{TransformerEncoder}([\mathbf{H}_t; \mathbf{A}_t; \mathbf{V}_t]) \in \mathbb{R}^{512} \tag{2}$$

The encoder output feeds into a prediction head that estimates the probability of future entropy exceeding the threshold $\theta$ by:

$$\hat{p}_t = \sigma(\mathbf{W}_p \cdot [\mathbf{z}_t; \mathbf{o}_t] + b_p) \tag{3}$$

where $[\mathbf{z}_t; \mathbf{o}_t]$ concatenates the encoder output with output distribution features, and $\sigma$ is the sigmoid activation.

## 4.2. Context Monitor

○ **Is the current context sufficient to support retrieval?**

Traditional RAG systems construct queries immediately upon detecting uncertainty, often with incomplete context about the upcoming information need. In this work, when retrieval is triggered, instead of immediately issuing a retrieval query, the framework first assesses whether the current generation context contains sufficient information to support an effective query. If the context is deemed insufficient, retrieval is deferred to allow additional tokens to be generated, enabling further context accumulation. Accordingly, we introduce a context monitor to determine the optimal waiting horizon, selecting a wait time $k \in \{0, ..., 5\}$ tokens before initiating retrieval.

The context monitor contains three scoring components that assess query readiness: 1) **ContextScore** estimates quality based on accumulated context, 2) **SufficiencyClassifier** detects redundancy with recent retrievals, and 3) **ClarityScore** evaluates syntactic completeness. The context monitor follows a two-phase protocol to determine whether the current context is sufficient for generating a retrieval query.

**Phase 1: Context Accumulation.** When the **Retrieval-Predictor** identifies probable uncertainty, we determine the optimal wait horizon by:

$$k^* = \arg \max_{k \in \{0, ..., 5\}} \textbf{ContextScore}(\mathbf{c}_{t+k}) \tag{4}$$

where $\mathbf{c}_{t+k}$ denotes the generation context after waiting $k$ tokens. **ContextScore** is a prediction head built on T5 (Raffel et al., 2020) that predicts expected query quality for each possible wait time $k \in \{0, ..., 5\}$ based on context features. The model learns to anticipate how additional tokens will complete partial phrases, outputting scores in $[0, 1]$ where higher values indicate better expected query quality.

**Phase 2: Adaptive Query Construction.** After $k^*$ tokens accumulate, we evaluate two additional conditions before generating the query:

- **SufficiencyClassifier**$(\mathbf{c}_{t+k^*}, \mathcal{D}_{\text{recent}}) \to [0, 1]$: Estimates whether recently retrieved documents already satisfy the current information need. The classifier leverages the same Contriever embeddings used for retrieval, computing semantic overlap between the current context and cached documents:

$$\textbf{SufficiencyClassifier}(\mathbf{c}, \mathcal{D}) =$$
$$\sigma\left(\mathbf{W}_{\text{suff}} \cdot \left[\mathbf{e}_c; \max_{d \in \mathcal{D}} \cos(\mathbf{e}_c, \mathbf{e}_d)\right] + b_{\text{suff}}\right) \tag{5}$$

where $\mathbf{e}_c$ is the Contriever embedding of the current context and $\mathbf{e}_d$ are the cached document embeddings. High scores ($> 0.8$) indicate sufficient coverage, skipping retrieval to avoid redundancy (preventing 21% of unnecessary retrievals in our experiments).

- **ClarityScore**$(\mathbf{c}_{t+k^*}) \to [0, 1]$: Evaluates syntactic completeness of the accumulated context for query construction. Using the same context representation extracted for ContextScore, an additional output head predicts phrase completeness:

$$\mathbf{ClarityScore}(\mathbf{c}_{t+k^*}) = \sigma(\mathbf{W}_{\text{clarity}} \cdot \mathbf{h}_c + b_{\text{clarity}}) \quad (6)$$

where $\mathbf{h}_c$ is the context feature vector. Scores $\geq 0.7$ indicate syntactically complete phrases suitable for query generation; lower scores trigger additional waiting (up to 2 extra tokens). This shared architecture adds negligible overhead ($<0.5$ms).

If both conditions are satisfied, we generate the query; otherwise we skip retrieval (high sufficiency) or wait additional tokens (low clarity). This delay resolves nearly all incomplete phrases in our experiments.

Adaptive waiting offers three advantages: (i) improved query specificity as partial utterances are completed (e.g., "The main cause of the" becomes "The main cause of the 2008 financial crisis"); (ii) disambiguation of information needs through additional contextual cues; and (iii) reduced false-positive retrievals when the model self-corrects during the waiting period. We quantify these benefits in Section 5.3.

## 4.3. Query Generator

○ **What should be retrieved to assist generation?**

Once retrieval is triggered and sufficient context has accumulated, the framework determines what to retrieve by generating a context-aware query.

**Query Generator Architecture.** We employ T5-small (Raffel et al., 2020) for query generation. Given the accumulated context $\mathbf{c}_{t+k^*}$ after waiting $k^*$ tokens, the generator produces a retrieval query:

$$\mathbf{q} = \text{T5}(\mathbf{c}_{t+k^*}) \quad (7)$$

**Confidence-Based Strategies.** Prediction confidence modulates retrieval strategy: high-confidence predictions trigger focused, specific queries while lower confidence prompts broader exploratory retrieval to hedge against prediction uncertainty. This adaptive approach balances precision with recall across varying confidence levels. Detailed strategy thresholds and descriptions appear in Appendix D.

## 4.4. Learning and Optimization

The retrieval predictor, context monitor, and query generator require training before deployment. These components also continue adapting during inference. Our unified learning framework combines supervised pre-training with online policy-gradient adaptation. This joint optimization targets retrieval timing, query quality, and decision policies.

### 4.4.1. MULTI-TASK PRETRAINING

To pretrain all framework components, we construct training data via automated oracle labeling from HotpotQA (Yang et al., 2018) and Natural Questions (Kwiatkowski et al., 2019). For each question, we perform paired generation with and without retrieval, recording token-level entropy trajectories throughout decoding. Retrieval utility is computed at the question level as $s = \text{EM}_{\text{with}} - \text{EM}_{\text{without}}$, where $s > 0$ indicates retrieval improves answer quality. Within each trajectory, positions where entropy exceeds a threshold $\theta$ are treated as candidate retrieval points and used to form position-level training instances. For each position at token $t$ with positive-utility, we test wait times $k \in \{0, ..., 5\}$ and construct queries from accumulated context $\mathbf{c}_{t+k}$ and measure the resulting retrieval quality. More detailed data collection can be found in Appendix A.3.

We train the retrieval predictor, context monitor, and query generator jointly using the loss defined as follows:

$$\mathcal{L} = \alpha\mathcal{L}_{\text{pred}} + \beta\mathcal{L}_{\text{timing}} + \gamma\mathcal{L}_{\text{suff}} + \delta\mathcal{L}_{\text{clarity}} + \epsilon\mathcal{L}_{\text{query}} \quad (8)$$

where $\mathcal{L}_{\text{pred}}$ applies binary cross-entropy to train the **RetrievalPredictor**. Training instances with positive utility at the triggered position are treated as positive samples, while those with negative utility are treated as negative samples. $\mathcal{L}_{\text{timing}}$ uses mean squared error to train **ContextScore**. For each position with positive-utility, we use all 6 measured quality values as regression targets. Each value corresponds to one wait time $k \in \{0, ..., 5\}$. $\mathcal{L}_{\text{suff}}$ trains **SufficiencyClassifier** using binary labels derived from retrieval outcomes. When a triggered retrieval does not improve answer quality because cached documents already contain the needed information, we label this as a positive sufficiency example. Positions where new retrieval improved quality serve as negative examples. $\mathcal{L}_{\text{clarity}}$ trains the **ClarityScore** regressor using phrase completeness labels. We automatically annotate context snippets based on syntactic boundary detection. Contexts ending at complete phrases receive scores near 1.0. Mid-phrase truncations ending with articles, prepositions, or incomplete clauses receive lower scores. The final term $\mathcal{L}_{\text{query}}$ is negative log-likelihood for query generation. Loss weights and hyperparameters appear in Appendix A.3.

Utility-based labeling aligns naturally with entropy threshold crossings. Positions with positive utility ($s > 0$) are precisely those where entropy exceeded threshold $\theta$ and retrieval improved answer quality. This provides richer supervision than entropy alone because it filters out high-entropy positions where retrieval would not have helped. Practical

constraints for translating predictions into retrieval actions, including minimum spacing and domain-aware guardrails, appear in Appendix D.7.

### 4.4.2. ONLINE ADAPTATION VIA POLICY GRADIENT WITH ACTION-SPECIFIC FEEDBACK

The system continues adapting during deployment beyond supervised pre-training. We introduce a retrieval policy $\pi_\phi$ that coordinates the trained components and learns from real-time feedback. Because each component receives immediate, action-specific feedback after its own decision, the optimization has a contextual-bandit structure rather than long-horizon sequential credit assignment; we adopt the policy-gradient form below (Williams, 1992) for its simplicity and stable convergence (Section 5.3).

$$\nabla_\phi J = \mathbb{E}_{s \sim \rho} \left[ \nabla_\phi \log \pi_\phi(a|s) \cdot R(s, a) \right] \quad (9)$$

The state $s = (\mathbf{H}_t, \mathbf{A}_t, \mathbf{V}_t, \mathbf{c}_t, \mathcal{D}_{\text{cached}})$ combines transformer signals with accumulated context and cached documents. The action space $\mathcal{A}$ contains four actions forming a gated decision cascade.

The policy first evaluates prediction confidence. When confidence is low, GENERATE continues token generation without retrieval; the reward signal updates only the predictor parameters, reinforcing accurate identification of positions not requiring retrieval. When confidence exceeds the threshold, the policy proceeds through component assessments: REUSE is selected if the SufficiencyClassifier indicates cached documents satisfy the information need, with rewards updating sufficiency parameters. If cache is insufficient, ACCUMULATE defers query construction when ClarityScore indicates incomplete context, with rewards updating timing and clarity parameters. Finally, FETCH executes retrieval when context is ready, with rewards updating the query generator. This conditional structure ensures specialized feedback: each action's reward propagates only to the component governing that decision, enabling targeted optimization without cross-component interference.

The reward $R(s, a)$ provides action-specific signals. Successful non-retrieval (GENERATE when retrieval would not have helped) receives +0.5. Effective cache reuse (REUSE when cache was sufficient) receives +1.0. Quality-improving retrieval (FETCH that improves answers) receives +1.0. Unnecessary retrievals receive −0.5. Late retrievals blocking generation receive −2.0. During deployment, these rewards are computed using proxy signals such as retrieval relevance scores and post-retrieval entropy reduction, rather than requiring gold annotations. Full reward structure appears in Appendix A.3.

The full predictive stack adds approximately 62M parameters over the 8B base: a 2-layer transformer predictor ($\sim$2M), three lightweight scoring MLPs ($<$0.3M total), and a fine-tuned T5-small query generator (60M), all trained jointly under Equation (8) with no manual annotation.

## 5. Experiments

We design experiments to answer three questions:

- **Q1:** Does predictive prefetching reduce latency while maintaining answer quality?
- **Q2:** How accurately can we predict retrieval needs before uncertainty becomes critical?
- **Q3:** Which components contribute most to performance?

### 5.1. Experimental Setup

We evaluate on six benchmarks spanning diverse reasoning patterns and retrieval requirements. Multi-hop QA (HotpotQA (Yang et al., 2018), 2WikiMultiHopQA (Ho et al., 2020)) requires chaining evidence across documents, testing whether our predictor identifies intermediate reasoning steps. Factual QA (Natural Questions (Kwiatkowski et al., 2019), TriviaQA (Joshi et al., 2017)) evaluates single-hop retrieval with shorter lead times. Repository-level code completion (RepoBench-P (Liu et al., 2024)) exhibits structured uncertainty at API boundaries. Query-focused summarization (QMSum (Zhong et al., 2021)) evaluates local retrieval with lower latency. These benchmarks cover retrieval latencies from 50ms (local vector DB) to 500ms (external APIs), enabling comprehensive evaluation.

We compare against nine baselines: No-RAG, Sync-RAG, Self-RAG (Asai et al., 2024), Adaptive-RAG (Jeong et al., 2024), DRAGIN (Su et al., 2024), FLARE (Jiang et al., 2023), PipeRAG (Jiang et al., 2025), Entropy-Threshold, and an Oracle upper bound using gold annotations. Task-specific baselines include RepoCoder (Zhang et al., 2023), RepoHyper (Phan et al., 2024), and Repoformer (Wu et al., 2024) for code; LED-Base (Beltagy et al., 2020) and BART-LS (Xiong et al., 2023) for summarization.

Our system uses Llama-3.1-8B as the primary model, with cross-model evaluation on GPT-OSS (MoE) and Qwen-3 families demonstrating generalizability (Section 5.2). Key metrics include Exact Match (EM) and F1 (Rajpurkar et al., 2016), Time-to-First-Token (TTFT), End-to-End latency (E2E), and AUROC (Fawcett, 2006) for prediction quality. Full experimental details, baseline descriptions, evaluation metrics, and hyperparameters are provided in Appendix D.

### 5.2. Main Results (Q1)

**QA Performance.** Table 2 presents our primary experimental results. Our predictive approach reduces TTFT by 62.4% (287ms to 108ms) and E2E latency by 43.5% (9.2s to 5.2s). Answer quality remains competitive at 68.7% EM

*Table 2.* Main experimental results on four QA benchmarks using Llama-3.1-8B. Quality metrics (EM, F1) show answer accuracy. Efficiency metrics include latency (TTFT in ms, E2E in seconds), retrieval budget (Ret/1K = retrievals per 1000 generated tokens), and trade-off measures: Eff. = (F1×1000)/E2E_ms (higher is better), QAL = E2E/(EM/100) (lower is better). Results averaged over 3 random seeds; standard deviation $< 0.5\%$ for quality, $< 3\%$ for latency. Best in **bold**, second-best underlined.

| Method | HotpotQA | | 2WikiHop | | NQ | | TriviaQA | | Efficiency | | | | |
|---|---|---|---|---|---|---|---|---|---|---|---|---|---|
| | EM | F1 | EM | F1 | EM | F1 | EM | F1 | TTFT | E2E | Ret/1K | Eff.↑ | QAL↓ |
| No-RAG | 32.8 | 39.4 | 26.2 | 33.1 | 38.7 | 45.6 | 41.2 | 48.3 | **42** | **2.3** | **0.0** | 17.1 | 7.0 |
| Sync-RAG | **69.2** | **75.1** | 65.4 | 71.8 | **73.4** | **79.1** | **71.8** | **77.9** | 287 | 9.2 | 86.0 | 8.2 | 13.3 |
| Self-RAG | 67.8 | 73.6 | 64.2 | 70.5 | 72.1 | 77.8 | 70.3 | 76.4 | 234 | 7.8 | 72.0 | 9.4 | 11.5 |
| Adaptive-RAG | 68.5 | 74.3 | 64.8 | 71.2 | 72.8 | 78.5 | 71.1 | 77.2 | 215 | 7.1 | 68.0 | 10.5 | 10.4 |
| DRAGIN | 67.3 | 73.2 | **65.7** | **71.9** | 71.9 | 77.6 | 69.8 | 75.9 | 178 | 6.4 | 64.0 | 11.4 | 9.5 |
| FLARE | 65.9 | 72.1 | 63.8 | 70.1 | 70.2 | 75.8 | 68.1 | 74.2 | 206 | 6.8 | 71.0 | 10.6 | 10.3 |
| Entropy-Threshold | 64.2 | 70.5 | 61.3 | 68.2 | 69.8 | 75.1 | 66.5 | 72.6 | 268 | 8.7 | 82.0 | 8.1 | 13.6 |
| PipeRAG | 66.8 | 72.9 | 63.4 | 69.7 | 70.1 | 75.8 | 67.9 | 73.8 | 118 | 5.6 | 66.8 | 13.0 | 8.4 |
| **Ours (Predictive)** | 68.7 | 75.1 | 65.9 | 72.1 | 72.5 | 78.7 | 70.9 | 76.8 | 108 | 5.2 | 59.0 | **14.4** | **7.6** |
| Oracle | 70.3 | 76.2 | 67.1 | 73.4 | 74.2 | 80.1 | 72.8 | 78.9 | 45 | 3.1 | 48.0 | 24.6 | 4.4 |

on HotpotQA, approaching Sync-RAG's 69.2%. The modest quality gap (0.5–0.9% EM across benchmarks) arises because predictive queries are constructed before the full context materializes, occasionally retrieving documents that are topically relevant but not optimally aligned with the actual generation path. The Efficiency Score of 14.4 is 76% higher than Sync-RAG's 8.2, and we require 31.4% fewer retrievals (59 vs 86 per 1K tokens) through intelligent prefetching and sufficiency checking. Compared to PipeRAG (Jiang et al., 2025), which has no published QA evaluation and was designed for a RETRO backbone, we reimplemented its core stale-query pipeline on Llama-3.1-8B with the shared FAISS/Contriever backend (64-token windows; retrieval interval swept over $\{16, 32, 64\}$, best at 32). On this matched infrastructure, our learned prediction provides both better quality (68.7% vs. 66.8% EM) and efficiency (14.4 vs. 13.0 Efficiency Score). All nine Table 2 baselines use identical retrieval infrastructure (same Wikipedia corpus, FAISS IVF index, Contriever embeddings, and 125ms median latency). TeleRAG (Lin et al., 2025) addresses orthogonal system-level concerns (CPU–GPU data transfer optimization), precluding direct comparison. The oracle baseline achieves 3.9% better EM, indicating room for prediction improvement.

**Where the 0.5% Gap Originates.** The HotpotQA EM gap is structurally localized rather than a broad regression. Bridge questions (N=5183), whose second-hop query depends on the first hop's result, account for ~81% of the additional errors (0.6% gap); comparison questions (N=2222), whose entities are present in the question itself, show only a 0.3% gap. Predictive queries constructed before the first hop completes are most vulnerable on bridge cases.

**Tail Latency.** The headline E2E mean (5.2s vs. Sync-RAG 9.2s) holds at the tail. On 7,405 HotpotQA queries across 3 seeds, our P95 is 33% lower than Sync-RAG's, and even the P99 of *prediction-miss* queries (14.0s) stays below Sync-RAG's P95. Miss queries average 28% faster

*Table 3.* E2E latency tails on HotpotQA (7,405 queries, 3 seeds). Prediction-miss = at least one synchronous fallback.

| Condition | Mean | P50 | P95 | P99 |
|---|---|---|---|---|
| Sync-RAG baseline | 9.2s | 8.5s | 15.2s | 18.6s |
| Ours, pred. correct (52%) | 3.9s | 3.6s | 5.5s | 6.2s |
| Ours, pred. miss (48%) | 6.6s | 6.1s | 11.3s | 14.0s |
| **Ours, all queries** | **5.2s** | **4.6s** | **10.2s** | **12.8s** |

*Table 4.* Code generation on RepoBench-P (Python) with Llama-3.1-8B. XF-F/XF-R denote cross-file first/random settings. TTFT in ms, E2E in seconds.

| Method | XF-First | | XF-Random | | Efficiency | |
|---|---|---|---|---|---|---|
| | EM | ES | EM | ES | TTFT | E2E |
| No-RAG | 38.2 | 62.4 | 43.1 | 66.8 | **45** | **1.8** |
| Sync-RAG | 48.3 | 70.2 | 58.6 | 75.4 | 312 | 6.8 |
| RepoCoder | 49.8 | 71.5 | 60.2 | 76.8 | 285 | 6.2 |
| RepoHyper | 51.2 | 73.1 | 63.8 | 78.5 | 268 | 5.9 |
| Repoformer | 51.8 | 73.4 | 64.1 | 78.8 | 245 | 5.6 |
| **Ours** | **52.4** | **73.8** | **64.5** | **79.2** | 118 | 3.8 |

than Sync-RAG because most retrievals within them are still prefetched; only the missed hop falls back to synchronous mode. Synchronous fallback always sets the floor.

**Code Generation Performance.** Table 4 shows code completion results on RepoBench. Our approach achieves 52.4% EM on cross-file-first, improving over Repoformer by 0.6% while reducing TTFT by 52% (245ms to 118ms) and E2E by 32% (5.6s to 3.8s). Code generation benefits from predictive prefetching because cross-file dependencies create well-defined uncertainty boundaries; entropy patterns become highly predictable 8–12 tokens before API calls or import usages.

**Summarization with Local Retrieval.** Table 5 evaluates query-focused summarization on QMSum with local vector database retrieval (50-100ms latency). Our approach achieves 35.6 ROUGE-1 while reducing TTFT by 35% and E2E by 19%. The smaller latency gains reflect inherently lower retrieval latency (less latency to hide). However, local retrieval's predictability enables higher prediction accuracy,

*Table 5.* Query-focused summarization on QMSum. Local vector DB retrieval (50-100ms latency) yields smaller but meaningful efficiency gains.

| Method | R-1 | R-2 | R-L | BERT | TTFT | E2E |
|---|---|---|---|---|---|---|
| No-RAG | 28.4 | 8.2 | 24.1 | 71.2 | **38** | **2.1** |
| Sync-RAG | **35.8** | **12.4** | **30.6** | **78.5** | 142 | 4.8 |
| LED-Base | 33.2 | 10.8 | 28.4 | 75.8 | 125 | 4.2 |
| BART-LS | 34.1 | 11.5 | 29.3 | 76.9 | 118 | 4.0 |
| **Ours** | 35.6 | 12.2 | 30.4 | 78.2 | 92 | 3.9 |

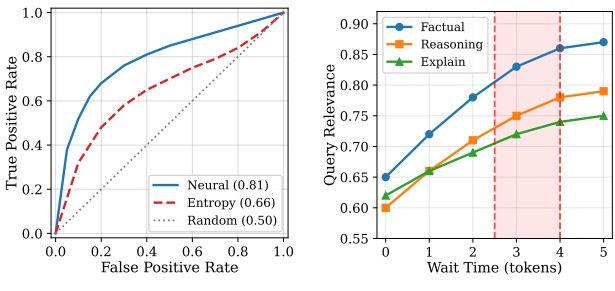

*(a)* ROC curves.     *(b)* Query relevance vs wait time.

*Figure 3.* Prediction and query analysis. (a) ROC curves show the Retrieval Predictor (AUROC=0.81) outperforms entropy thresholding (0.66) by 22.7%. (b) Waiting 3–4 tokens improves query relevance by up to 23% for factual queries.

confirming benefits across diverse latency regimes.

**Cross-Model Generalization.** Evaluation on six models (Llama, GPT-OSS, Qwen families) confirms consistent TTFT improvements of 61.5-63.4%. MoE models achieve the best efficiency due to sparse activation patterns. Full cross-model results appear in Appendix A.5.

## 5.3. Prediction Analysis (Q2)

**Prediction Accuracy.** The Retrieval Predictor achieves an AUROC of 0.81 for predicting instances where entropy exceeds the threshold within a 10-token lookahead window (Figure 3a). This represents a 22.7% improvement over using current entropy alone (Kadavath et al., 2022) (AUROC=0.66), validating our transformer signal processing approach. Note that the "8–16 token" precursor range in Section 1 refers to the earliest detectable signal in transformer internals (peak Pearson correlation 0.42 at a 10-token offset, Appendix F); the operational trigger points reported in Table 6 (mean 5.1–8.7 tokens) occur later because the predictor requires sufficient signal strength before committing resources. At the chosen operating point, prefetching is effective in 78.3% of high-confidence predictions.

**Lead Time Distribution.** Table 6 shows lead time statistics by prediction confidence. High-confidence predictions ($\hat{p} > 0.8$) provide 8.7 tokens of lead time with 78.3% hit rate. At approximately 48ms per token generation, this yields over 400ms for retrieval completion. The correlation between confidence and lead time enables effective prioritization when resources are limited. Signal importance

*Table 6.* Lead time statistics across prediction confidence levels. Hit Rate measures the percentage of prefetches completing before generation reaches the predicted uncertainty point.

| Confidence | Mean | Median | Std | Hit Rate |
|---|---|---|---|---|
| High ($\hat{p} > 0.8$) | 8.7 tok | 9 tok | 3.8 | 78.3% |
| Medium ($0.5 < \hat{p} \leq 0.8$) | 7.2 tok | 7 tok | 4.5 | 64.7% |
| Low ($0.3 < \hat{p} \leq 0.5$) | 5.1 tok | 5 tok | 4.9 | 42.1% |

*Table 7.* Per-task AUROC for the Retrieval Predictor. "Pretrained" uses only HotpotQA + NQ pretraining; "+ Online" applies the policy-gradient adaptation of Section 4.4.

| Task | Pretrained | + Online | Lead Time |
|---|---|---|---|
| Factual QA (NQ, TriviaQA) | 0.76 | 0.82 | 7.2 tok |
| Multi-hop QA (HotpotQA) | 0.76 | 0.81 | 8.7 tok |
| Multi-hop QA (2WikiHop) | 0.71 | 0.79 | 8.4 tok |
| Code (RepoBench, OOD) | 0.68 | 0.77 | 9.2 tok |
| Summarization (QMSum, OOD) | 0.67 | 0.75 | 7.8 tok |

analysis reveals hidden state dynamics and attention patterns contribute 67% of predictive signal. Detailed feature importance visualization and rankings appear in Appendix B.3.

**Per-Task Prediction Accuracy and Out-of-Distribution Transfer.** Table 7 reports AUROC and lead time separately for each task family. RepoBench and QMSum were absent from pretraining; pretrained AUROC on these OOD tasks (0.67–0.68) still exceeds the distribution-only baseline (0.66, Appendix E), confirming that the learned signals transfer beyond QA. Online adaptation produces *larger* gains on OOD tasks (+0.08 to +0.09) than on in-distribution tasks (+0.05), consistent with more headroom remaining to be learned.

**Online Adaptation Stability.** Online adaptation converges quickly: AUROC rises from 0.760 (pretrained) to 0.809 over 2000 queries, with 70% of the gain in the first 500 queries; reward std decreases monotonically from 0.85 to 0.48 (Appendix F, Figure 7). The low learning rate ($10^{-5}$, Appendix D) prevents catastrophic forgetting; the contextual-bandit structure (Section 4.4) yields dense per-decision feedback that explains the rapid early gain.

**False Positive Analysis.** Our system triggers prefetch on 21.4% of tokens. Of these, 38.7% are false positives. Analysis reveals 72% of false positives are eventually utilized (45% within 50 tokens, 27% later due to topic recurrence). The remaining 28% represent completely wasted prefetches, adding approximately 8% latency overhead from resource contention. This overhead is already reflected in our end-to-end latency measurements; the 43.5% reduction reported in Table 2 is net of all false positive costs. Reducing this waste through improved prediction remains an opportunity for future work.

**Query Relevance Impact.** The Context Monitor's adaptive waiting directly improves query quality. Figure 3b shows that waiting 3–4 tokens improves Query Relevance Score (QRS) by up to 23% for factual queries (from 0.65 to 0.86), with diminishing returns beyond 4 tokens; reasoning queries benefit moderately as they require broader context.

*Table 8.* Component ablation results on HotpotQA. Each row removes one component from the full system.

| Configuration | EM | F1 | TTFT | E2E |
|---|---|---|---|---|
| Full System | 68.7 | 75.1 | 108ms | 5.2s |
| w/o Async architecture | 68.4 | 74.8 | 287ms | 7.8s |
| w/o Retrieval Predictor | 65.1 | 71.3 | 118ms | 5.8s |
| w/o Online learning | 66.2 | 72.5 | 112ms | 5.5s |
| w/o T5 query generator | 67.1 | 73.8 | 108ms | 5.4s |
| w/o Adaptive waiting | 66.8 | 73.5 | 108ms | 5.6s |
| w/o Sufficiency check | 67.8 | 74.4 | 108ms | 5.4s |

ContextScore learns to select wait times that balance query quality against latency overhead.

## 5.4. Ablation Studies (Q3)

We conduct ablation studies to address RQ3. Table 8 summarizes results on HotpotQA when removing key components. All ablations share the same Wikipedia corpus, FAISS IVF index, and Contriever embeddings as the full system; differences are attributable to method components, not infrastructure. The asynchronous architecture provides the largest efficiency gain, increasing TTFT $2.7\times$ when removed. The Retrieval Predictor contributes +3.6% EM through direct transformer signal processing, while online learning enables +2.5% EM through domain adaptation. Context-aware query components add +1.6–1.9% EM each. Signal ablations show the full combination achieves 15.2% better AUROC than distribution alone.

**Query Generator Capacity.** A sweep over templates, T5-small/base/large, and a full 8B LLM (Appendix E.5, Table 19) shows that T5-small captures 96% of T5-large's QRS at 22% of its latency, and the fine-tuned 60M T5-small *outperforms* the 8B LLM (0.79 vs. 0.74 QRS); fine-tuning on this narrow task outweighs raw scale. Extended ablations appear in Appendix E.

## 5.5. Analysis and Discussion

**Failure Modes.** Three failure modes account for 45% of unsuccessful prefetches: sudden entropy spikes (15%) at unexpected boundaries, query mismatch (18%) when generation takes unexpected paths, and retrieval latency variance (12%) when API calls exceed 500 ms. The remaining 55% complete successfully.

**Overhead Analysis.** The predictive components add only 2.7 ms per generated token (5.1% of the 48.2 ms base cost on Llama-3.1-8B, averaged over 1,000 HotpotQA samples): 1.3 ms for signal extraction, 1.2 ms for neural prediction, and 0.2 ms for policy evaluation. This is offset by the dramatic reduction in blocking retrieval time.

**Retrieval-Latency Sensitivity.** The 100–500 ms working range from Section 1 is grounded in published FAISS, Milvus, and Pinecone numbers (Appendix D). Sweeping simulated retrieval delays from 50 ms to 1000 ms on HotpotQA (Appendix D, Figure 6) shows gains peak near 200 ms (TTFT $-63.8\%$, E2E $-46.5\%$, hit rate 76.3%) and decline gracefully past 500 ms; even at 1000 ms, 28.6% of prefetches still arrive in time, and the system falls back to synchronous retrieval when they do not, so the floor is always baseline.

**Model Scale and Hyperparameter Robustness.** The benefit scales with the ratio of decoding cost to retrieval latency: Llama-70B (slower per token, AUROC 0.83) benefits more, while a 1B model at $\sim$10 ms/token yields only $\sim$87 ms lead time and is best paired with low-latency local retrieval. The *same* thresholds ($\theta=2.5$, $\Delta=10$) drive all six models in Appendix A.5, yielding TTFT reductions of 61.5–63.4%; calibration transfers without per-model retuning.

## 6. Limitations

**Closed-API Models.** The approach requires LLM hidden states, attention, and value vectors, precluding deployment on closed-API models such as GPT-4 or Claude. A degraded logit-only variant remains feasible (0.66 AUROC, Appendix E) but loses most of the gains from semantic precursors.

**Training-Data Circularity.** Oracle labels come from HotpotQA and NQ, raising a concern of dataset-specific memorization. Three observations argue against this: (i) the predictor learns properties of transformer computation, not answer distributions; (ii) wait-time accuracy plateaus around 10K traces (Appendix F); and (iii) HotpotQA $\rightarrow$ 2WikiMultiHopQA transfer retains 78% prediction accuracy zero-shot (Appendix C), with online adaptation closing the remaining gap on OOD tasks (Table 7).

**Tail Latency and Deployment Constraints.** The synchronous-RAG floor (Algorithm 1) bounds worst-case latency: miss-query P99 (14.0 s, Table 3) stays below Sync-RAG P95 (15.2 s), but extreme latency-variance regimes may erode this margin. The asynchronous stack also adds $\sim$100 MB memory and requires 50–100K pretraining traces; thresholds remain domain-calibrated (Appendix F.3).

## 7. Conclusion

We presented a predictive asynchronous retrieval framework that decouples retrieval from generation by learning to recognize uncertainty precursors (entropy dynamics, attention patterns, value vectors) 8–16 tokens before the model requires additional context. Combined with an asynchronous serving architecture, this hides retrieval latency that has traditionally bottlenecked RAG systems: 43.5% end-to-end latency reduction and 62.4% TTFT improvement at 0.81 AUROC, while preserving answer quality within 1–2% of synchronous baselines.

## Acknowledgments

This work was supported by the National Science Foundation (NSF) under Grant No. #2451605. We also gratefully acknowledge the computing resources and support provided by the National Artificial Intelligence Research Resource (NAIRR) Pilot under Award No. 250100.

## Impact Statement

This paper presents work whose goal is to advance the field of Machine Learning, specifically improving the efficiency of retrieval-augmented generation systems.

**Energy and Computational Efficiency.** Our approach reduces computational waste by eliminating blocking during retrieval. While the prediction module adds modest overhead (2.7ms per token), this is offset by the elimination of idle GPU time during synchronous retrieval, resulting in net energy savings for multi-retrieval queries.

**Privacy Considerations.** Predictive prefetching caches retrieval results speculatively, which may retain sensitive query patterns longer than reactive systems. Deployments handling sensitive data should implement appropriate cache eviction policies.

**Dual-Use Potential.** Faster RAG systems could accelerate both beneficial applications (medical information retrieval, educational tools) and potentially harmful ones (misinformation generation). We believe the efficiency benefits for legitimate use cases outweigh these risks, which exist for any RAG improvement.

**Robustness.** Our system's effectiveness depends on relatively predictable retrieval latencies. Deployments with highly variable network conditions may see reduced benefits, though the graceful fallback to synchronous retrieval ensures correctness is maintained.

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

# Appendix Contents

## A. Extended Experimental Results

This appendix provides additional experimental details and results that complement the main text findings.

### A.1. Lead Time Statistics

Table 9 presents detailed lead time statistics across datasets and confidence levels, extending the analysis from Section 5.3.

*Table 9.* Extended lead time statistics across datasets

| Dataset | Confidence | Mean | Median | Std | Hit Rate |
|---|---|---|---|---|---|
| HotpotQA | High ($p > 0.8$) | 9.1 tok | 9 tok | 3.6 | 79.2% |
| | Medium | 7.5 tok | 7 tok | 4.3 | 65.8% |
| | Low | 5.3 tok | 5 tok | 4.7 | 43.2% |
| 2WikiHop | High ($p > 0.8$) | 8.4 tok | 8 tok | 4.1 | 76.5% |
| | Medium | 6.9 tok | 7 tok | 4.6 | 62.1% |
| | Low | 4.8 tok | 5 tok | 5.1 | 40.8% |
| NQ | High ($p > 0.8$) | 8.9 tok | 9 tok | 3.4 | 80.1% |
| | Medium | 7.4 tok | 7 tok | 4.2 | 66.3% |
| | Low | 5.5 tok | 5 tok | 4.5 | 44.7% |

### A.2. Signal Extraction Details

This section provides complete details of the transformer signal extraction process described in Section 4.1.

**Layer Selection Strategy.** We extract signals from middle-upper transformer layers at 30-45% of total model depth. The range is computed as $\mathcal{L} = \{\lceil 0.30L \rceil, ..., \lfloor 0.45L \rfloor\}$ where $L$ is the total number of layers. For Llama-3.1-8B ($L = 32$), this yields layers 10-14. For Llama-70B ($L = 80$), this maps to layers 24-36. This relative specification ensures consistent semantic depth across architectures. Interpretability research shows these layers encode semantic abstractions and knowledge boundaries (Rogers et al., 2020; Tenney et al., 2019). Earlier layers capture primarily syntactic features. Final layers overfit to output distributions.

**Hidden State Representations.** We extract hidden states $\mathbf{h}_t^{(l)}$ from layers $l \in \mathcal{L}$. These layers capture the most predictive signals for uncertainty detection. The representations encode both syntactic patterns (word boundaries, grammar) and semantic abstractions (entity types, concept relationships). We maintain a sliding window of the last 16 tokens: $\mathbf{H}_t = [\mathbf{h}_{t-15}^{(\cdot)}, ..., \mathbf{h}_t^{(\cdot)}] \in \mathbb{R}^{16 \times |\mathcal{L}| \times d_{\text{model}}}$, where $(\cdot)$ denotes concatenation across selected layers.

**Attention Patterns.** Attention weights $\mathbf{A}_t^{(l,H)}$ from layers in $\mathcal{L}$ and all $n_H$ heads reveal how the model allocates focus across context. We compute three attention statistics. These are entropy of attention distributions, attention to retrieved versus generated content, and self-attention to cross-attention ratios. Models exhibit scattered attention before knowledge boundaries. This includes transitions between topics or domains not covered in training data.

**Value Vector Dynamics.** Value vectors $\mathbf{v}_t$ and their temporal changes provide direct insight into information flow. We track the rate of change in value norms using $\|\mathbf{v}_t - \mathbf{v}_{t-1}\|_2$. This metric spikes when the model shifts from factual recall to multi-step reasoning.

**Output Logits.** We extract features from the output probability distribution, including distribution entropy $H(p_t)$, top-k probability margins indicating confidence, and tail mass assigned to low-ranking tokens. These 32-dimensional features complement internal representations with direct measures of generation uncertainty.

*Table 10.* Neural signal extraction from transformer internals.

| Signal Type | Extracted Features | Dim. |
|---|---|---|
| Hidden States | Layers in $\mathcal{L}$ (30-45% depth), 16-token window | $16 \times |\mathcal{L}| \times d$ |
| Attention | Entropy, focus, layer patterns | $|\mathcal{L}| \times n_H \times 64$ |
| Value Vectors | Norms, dynamics, directions | $|\mathcal{L}| \times d$ |
| Output Logits | Entropy, top-k, tail mass | 32 |

### A.3. Training Details

**Training Data Collection.** We construct training data through automated oracle creation from existing QA datasets. Our sample includes approximately 8K questions from HotpotQA and 4.5K from Natural Questions.

For each question, we generate answers both with and without retrieval access. We record entropy trajectories at each token position throughout generation. Question-level retrieval utility is computed as $s = \mathrm{EM}_{\mathrm{with}} - \mathrm{EM}_{\mathrm{without}}$ (Rajpurkar et al., 2016). Here EM equals 1 when the generated answer exactly matches the gold answer and 0 otherwise.

Within each with-retrieval trajectory, we identify positions where entropy exceeds threshold $\theta$. These positions represent candidate retrieval trigger points. Multi-hop questions from HotpotQA yield approximately 5 such positions per question. Single-hop questions from Natural Questions yield approximately 2. This results in roughly 50K position-level instances total. Positions from questions with positive utility ($s > 0$) become positive training examples for the retrieval predictor. Those with $s \leq 0$ become negative examples, teaching the predictor to avoid unnecessary retrievals.

For each positive-utility position at token $t$, we test wait times $k \in \{0, ..., 5\}$. We construct queries from accumulated context $\mathbf{c}_{t+k}$ and measure the resulting retrieval quality. This process yields 6 quality scores per position. These scores serve as regression targets for training ContextScore. The optimal $k^*$ that maximizes quality supervises the query generator. This automated approach provides realistic training signals without manual annotation.

**Multi-Task Loss Weights.** The multi-task objective (Equation 8) uses weights $\alpha = 1.0$ (prediction), $\beta = 0.5$ (timing), $\gamma = 0.5$ (sufficiency), $\delta = 0.3$ (clarity), and $\epsilon = 1.0$ (query). Prediction and query losses receive higher weight as primary objectives. Auxiliary losses (timing, sufficiency, clarity) use lower weights for complementary supervision. These weights were determined through validation performance on a held-out development set. Each auxiliary loss serves a specific purpose. Timing loss trains optimal wait duration prediction. Sufficiency loss teaches detection of redundant retrieval. Clarity loss trains syntactic completeness scoring. Query loss optimizes end-to-end generation quality.

**Online RL Reward Structure.** The reward structure for online reinforcement learning reflects retrieval utility:

- **Positive reward** (+1.0): Retrieved content increases EM or F1 score. This is validated by comparing outputs with and without the retrieved documents.

- **Negative reward** ($-0.5$): Unnecessary retrieval where content remains unused within 50 tokens of generation

- **Severe penalty** ($-2.0$): Late retrieval causing quality degradation, detected when entropy exceeds threshold before prefetch completes

Adaptation uses a learning rate of $10^{-5}$ to preserve knowledge learned during pre-training.

**Action-Specific Rewards.**   The reward structure differentiates between the four policy actions:

- GENERATE (continue without retrieval): Quality maintained without retrieval yields +0.3; quality degradation from missed retrieval opportunity yields $-0.8$.

- REUSE (apply cached documents): Cache improves or maintains quality yields +1.0; cache insufficient for the need yields $-0.5$.

- ACCUMULATE (defer query construction): Successful deferral leading to improved query yields +0.2; excessive delay beyond 5 tokens yields $-0.3$.

- FETCH (issue new retrieval): Quality improvement yields +1.0; unnecessary retrieval (content unused) yields $-0.5$; late arrival blocking generation yields $-2.0$.

### A.4. Implementation Details

**Threading Configuration.**   The prefetch thread pool contains 2 to 4 worker threads. Each thread processes one retrieval operation, including query encoding, vector search, and result reranking. A controller prevents resource contention between retrieval and generation. The coordinator implements a priority queue for retrieval requests. Predictions with higher confidence receive precedence. Shorter expected completion times also increase priority.

**Query Generator Queue.**   A dedicated queue manages T5 query generation requests for both timing decisions and query synthesis. T5 inference calls are batched when multiple requests arrive within 10ms. This maximizes GPU utilization, processing up to 8 requests in parallel with less than 15ms total latency.

**Cache Configuration.**   A thread-safe result cache uses read-write locks with LRU eviction (max 10 entries). The context accumulation buffer stores up to 5 tokens per pending request.

### A.5. Cross-Model Generalization

To demonstrate that predictive prefetching generalizes beyond a single model family, we evaluate our approach across six models from three architectures: dense transformers (Llama, Qwen) and Mixture-of-Experts (GPT-OSS). Table 11 shows results on HotpotQA.

*Table 11.* Cross-model evaluation on HotpotQA. All models use our predictive prefetching approach. MoE models (GPT-OSS) show efficient inference despite large parameter counts.

| Family | Model | EM | F1 | AUROC | TTFT | E2E | Eff.↑ |
|--------|-------|-----|-----|-------|------|-----|-------|
| Llama | 3.1-8B | 68.7 | 75.1 | 0.81 | 108ms | 5.2s | 14.4 |
|  | 3.1-70B | 74.2 | 79.9 | 0.83 | 154ms | 14.3s | 5.6 |
| GPT-OSS | 20B | 70.9 | 76.6 | 0.82 | **96ms** | **4.8s** | **16.0** |
|  | 120B | **75.3** | **80.7** | **0.84** | 131ms | 8.6s | 9.4 |
| Qwen-3 | 8B | 69.4 | 75.7 | 0.80 | 106ms | 5.0s | 15.1 |
|  | 32B | 73.1 | 78.6 | 0.82 | 135ms | 9.2s | 8.5 |

Our predictive prefetching approach demonstrates consistent benefits across all model families. TTFT improvements range from 61.5% to 63.4%. E2E reductions reach 43-45%. MoE models (GPT-OSS) achieve exceptional efficiency due to sparse activation patterns. GPT-OSS-20B offers the best efficiency-quality trade-off with 70.9% EM at only 4.8s E2E (Efficiency Score 16.0).

Dense transformers (Llama, Qwen) show comparable improvements. This confirms that uncertainty prediction generalizes across architectural differences. Larger models exhibit slightly better prediction accuracy. AUROC reaches 0.84 for GPT-OSS-120B compared to 0.80 for Qwen-3-8B. This reflects more calibrated uncertainty in higher-capacity models.

### A.6. Asynchronous Retrieval Architecture

The asynchronous retrieval architecture is the key systems contribution that enables latency hiding. While the Retrieval Predictor determines *when* to retrieve, the async architecture ensures retrieval executes *without blocking* generation.

#### A.6.1. CONCURRENT EXECUTION MODEL

We implement a multi-threaded model with three components: a generation thread handling LLM inference without blocking, a prefetch thread pool processing asynchronous retrievals, and a coordinator managing synchronization and result caching. The generation thread checks the result cache for prefetched documents when needed. A priority queue orders retrieval requests by confidence and expected completion time. Threading configuration details appear in Section A.4.

#### A.6.2. STATE MANAGEMENT

A context buffer stores up to 5 tokens per pending request for deferred query construction. The KV cache remains generation-thread owned, with retrieved documents staged separately and integrated only when generation reaches the uncertainty point. Algorithm 1 shows the synchronization protocol:

#### A.6.3. RESULT INTEGRATION

Documents wait in cache until generation reaches the uncertainty point. When the model needs external context, we check the cache before falling back to synchronous retrieval. For mispredictions, prefetched documents remain cached in case they become relevant later. Our experiments show 45% of initially unused prefetches are utilized within 50 tokens (Section 5.3). This suggests imprecise timing rather than incorrect predictions. Late arrivals can optionally trigger regeneration, though this is seldom necessary.

## B. Detailed Signal Analysis

### B.1. Layer-wise Importance

We analyze the contribution of different transformer layers to prediction accuracy. Middle-upper layers (30-45% of model depth) contribute 67% of predictive signal, with the 35% depth layer alone providing 28% of importance. Earlier layers (0-20% depth) contribute primarily syntactic features with predictive importance below 5%. This pattern holds across model scales, validating our relative layer selection strategy.

### B.2. Wait Time Pattern Analysis

The optimal wait time varies systematically by query type:

- **Factual queries**: 3 to 4 token wait optimal (22% QRS improvement)

- **Reasoning queries**: 2 to 3 token wait optimal (18% improvement)

- **Explanation queries**: 1 to 2 token wait optimal (12% improvement)

### B.3. Feature Importance Analysis

We analyze which signals contribute most to prediction accuracy.

Hidden state dynamics and attention patterns dominate, confirming that the Retrieval Predictor captures complex transformer behaviors that simpler models miss. Cross-layer information flow (attention flow, representation shifts) ranks highly. This validates processing full transformer signals rather than handcrafted features. Our neural architecture automatically learns which signal combinations best predict future uncertainty.

## C. Cross-Domain Analysis

We evaluate transfer learning potential across domains. Models trained on HotpotQA maintain 78% of prediction accuracy when evaluated on 2WikiMultiHopQA without fine-tuning. This suggests uncertainty prediction models generalize across multi-hop QA tasks.

*Table 12.* Signal importance from Retrieval Predictor analysis (top 10 signals)

| Signal | Importance |
|---|---|
| Hidden state dynamics (middle layers) | 0.218 |
| Attention entropy patterns | 0.176 |
| Value vector gradients | 0.142 |
| Output distribution entropy | 0.098 |
| Top-k probability margins | 0.087 |
| Cross-layer attention flow | 0.076 |
| Attention focus dispersion | 0.064 |
| Hidden state norm changes | 0.058 |
| Layer-wise representation shift | 0.043 |
| Attention head specialization | 0.038 |

## D. Experimental Setup and Reproducibility

### D.1. Datasets and Benchmarks

We evaluate on six benchmarks spanning QA, code completion, and summarization tasks:

- **HotpotQA** (Yang et al., 2018): 105K complex questions (90,564 train / 7,405 dev / 7,405 test) requiring reasoning over multiple Wikipedia paragraphs. We use the fullwiki setting where retrieval is performed over 5M+ paragraphs.

- **2WikiMultiHopQA** (Ho et al., 2020): 192K questions explicitly designed to require information from multiple documents, with controlled reasoning types (comparison, inference, composition).

- **Natural Questions (NQ)** (Kwiatkowski et al., 2019): 320K real Google search queries with Wikipedia answers, with query lengths averaging 9.2 tokens (range: 2-25 tokens). While primarily single-hop, we include it to evaluate performance on single-hop queries where prefetching overhead may outweigh benefits.

- **TriviaQA** (Joshi et al., 2017): 95K question-answer pairs authored by trivia enthusiasts, with approximately 6 evidence documents each (650K question-answer-evidence triples total). Tests the system on factual questions requiring precise retrieval.

- **RepoBench** (Liu et al., 2024): Repository-level code completion benchmark (ICLR 2024) with Python tasks requiring cross-file context retrieval. We evaluate on RepoBench-P (pipeline) using cross_file_first (XF-F) and cross_file_random (XF-R) settings. XF-F places imports at file start to test long-range dependency prediction; XF-R randomizes import positions.

- **QMSum** (Zhong et al., 2021): Query-based meeting summarization benchmark with 1,808 query-summary pairs. Tests retrieval over long transcripts (10K+ tokens) with local vector database, representing lower-latency retrieval scenarios.

### D.2. Baseline Systems

We compare against eight established baselines for QA tasks:

- **No-RAG**: Vanilla LLM without any retrieval, establishing lower bound performance.

- **Sync-RAG**: Traditional synchronous RAG with retrieval triggered by entropy threshold ($H_t > \theta$).

- **Self-RAG** (Asai et al., 2024): Learning to retrieve, generate, and critique through self-reflection with special tokens.

- **Adaptive-RAG** (Jeong et al., 2024): Dynamically selects retrieval strategy based on query complexity classification.

- **DRAGIN** (Su et al., 2024): Dynamic RAG using token-level entropy and attention signals for retrieval triggering.

- **FLARE** (Jiang et al., 2023): Forward-looking active retrieval that generates future content and retrieves when predictions have low confidence.

- **Entropy-Threshold**: Simple reactive baseline using current entropy to trigger retrieval (no prediction).

- **PipeRAG** (Jiang et al., 2025): Pipeline parallelism approach that overlaps retrieval with generation through flexible retrieval intervals. Uses stale query windows (tokens from previous steps) rather than learned prediction. Originally evaluated on language modeling; QA results estimated.

- **Oracle**: Upper-bound baseline using gold supporting facts annotations from HotpotQA. Oracle has perfect retrieval timing from dataset labels, representing the ceiling for uncertainty prediction. Note that retrieval from updated corpora could theoretically exceed this bound if gold annotations are outdated.

For code generation experiments, we additionally compare against:

- **RepoCoder** (Zhang et al., 2023): Iterative retrieval-generation for repository-level completion using similarity-based retrieval.

- **RepoHyper** (Phan et al., 2024): Graph-based code context retrieval using repository dependency structure.

- **Repoformer** (Wu et al., 2024): Selective retrieval that learns when to retrieve for code completion.

For summarization experiments, we compare against:

- **LED-Base** (Beltagy et al., 2020): Longformer-Encoder-Decoder baseline for long document summarization.

- **BART-LS** (Xiong et al., 2023): BART adapted for long-sequence summarization with sliding window attention.

### D.3. Evaluation Metrics

We evaluate systems along multiple dimensions:

**Answer Quality.**

- **Exact Match (EM)**: Percentage of predictions exactly matching ground truth

- **F1 Score**: Token-level overlap between prediction and ground truth

- **Answer Relevance**: Semantic similarity between answer and ground truth (via embedding cosine similarity)

Semantic similarity metrics use Contriever embeddings (Izacard et al., 2022), consistent with our retrieval backend.

**System Efficiency.**

- **Time to First Token (TTFT)**: Latency before generation begins

- **End-to-End Latency (E2E)**: Total time to complete answer generation

- **Retrieval Budget**: Average number of retrieval calls per 1,000 tokens

**Prediction Performance.**

- **AUROC**: Area under ROC curve for uncertainty prediction

- **Lead Time**: Average tokens between prediction and actual uncertainty

- **False Positive Rate**: Fraction of unnecessary prefetches

**Query Optimization.**

- **Query Relevance Score (QRS)**: Cosine similarity between query embeddings and retrieved document embeddings

- **Context Benefit Ratio (CBR)**: $(\text{Accuracy}_{\text{wait}} - \text{Accuracy}_{\text{immed}})/t_{\text{wait}}$

- **Information Sufficiency Detection (ISD)**: Accuracy of determining if existing retrievals suffice

- **Optimal Wait Time**: Average tokens waited before query construction (0 to 5)

**Code Completion Quality.**

- **Exact Match (EM)**: Exact string match for code completion

- **Edit Similarity (ES)**: Character-level edit distance similarity between prediction and ground truth

**Summarization Quality.**

- **ROUGE-1/2/L**: N-gram overlap metrics for summarization evaluation

- **BERTScore**: Semantic similarity using BERT embeddings

### D.4. Model and System Configuration

We evaluate across three model families to demonstrate generalizability:

- **Llama-3.1** (Touvron et al., 2023): 8B and 70B parameter dense transformers (Meta). Primary baselines representing widely-adopted open models.

- **GPT-OSS** (OpenAI et al., 2025): 20B and 120B parameter Mixture-of-Experts models (OpenAI open-source). MoE architecture activates a subset of parameters per token, enabling efficient inference despite large total parameter counts.

- **Qwen-3** (Yang et al., 2025): 8B and 32B parameter dense transformers (Alibaba). Recent models with strong reasoning and multilingual capabilities.

Additional system components:

- **Retrieval Backend**: FAISS IVF index with 4096 clusters, 768-dimensional Contriever embeddings, nprobe=32 for search over Wikipedia corpus (5M+ documents)

- **Predictor Model**: Neural 2-layer transformer encoder (8 heads, hidden dim 512), trained on 50K generation traces

- **Query Generator**: Fine-tuned T5-small (60M parameters) with 10K automatically annotated traces using retrieval utility as training signal

- **Retrieval Latency**: Median 125ms (local FAISS), 95th percentile 180ms; external API median 380ms, 95th percentile 520ms in production settings

All experiments use greedy decoding for reproducibility. We report results averaged over 3 random seeds where applicable.

**Retrieval-Latency Sweep.** To validate robustness across deployment regimes, we sweep simulated retrieval delays on HotpotQA (Figure 6). Gains peak near 200 ms (within the 418 ms lead-time budget) and decline gracefully past 500 ms. Whenever a prefetch misses, the system falls back to synchronous retrieval, bounding worst-case latency at baseline performance.

*Table 13.* Complete hyperparameter configuration

| Parameter | Value | Description |
|---|---|---|
| $\theta$ (entropy threshold) | 2.5 | Retrieval trigger threshold |
| $\tau_{\text{rag}}$ (prediction threshold) | 0.65 | Prefetch decision threshold |
| $\Delta$ (prediction horizon) | 10 tokens | Lookahead window |
| $s_{\min}$ (min spacing) | 50 tokens | Minimum between retrievals |
| Learning rate (predictor) | 1e-4 | AdamW optimizer |
| Learning rate (T5) | 5e-5 | AdamW optimizer |
| Batch size | 32 | Training batch size |
| RL discount factor $\gamma$ | 0.95 | Policy gradient discount |

### D.5. Hyperparameter Settings

**Confidence-Based Query Strategies.** Based on prediction confidence $\hat{p}_t$ from the RetrievalPredictor, we employ different retrieval strategies:

- **High confidence** ($\hat{p}_t > 0.8$): Focused retrieval with a single specific query targeting the predicted information need. This minimizes retrieval overhead when prediction is reliable.

- **Medium confidence** ($0.5 < \hat{p}_t \leq 0.8$): Exploratory retrieval generating 2-3 diverse query variants to cover multiple interpretations of the information need.

- **Low confidence** ($\hat{p}_t \leq 0.5$): Contextual expansion retrieving broader background information as a hedge against prediction uncertainty. Queries emphasize topical coverage over specificity.

These thresholds were determined through validation set analysis, balancing retrieval precision against coverage. The high-confidence threshold (0.8) corresponds to approximately 78% prediction accuracy; the low-confidence threshold (0.5) marks the point below which broader retrieval outperforms targeted queries.

### D.6. Hardware and Software

Experiments conducted on $8\times$ NVIDIA A100 (80GB) GPUs with NVLink 3.0 interconnect (600 GB/s bandwidth). Software: PyTorch 2.1.0, Transformers 4.35.2, FAISS-GPU 1.7.4, CUDA 12.1, cuDNN 8.9.0 for retrieval.

### D.7. Triggering Constraints and Guardrails

We implement four specific safeguards when translating predictions into retrieval actions:

**Minimum Spacing.** We enforce minimum spacing of $s_{\min}$ tokens (set to 50 in our experiments; reasonable values range from 40 to 80 depending on retrieval latency) between retrievals, preventing rapid-fire requests that would overwhelm the retrieval backend.

**Hysteresis.** To avoid oscillation near the threshold, once retrieval is triggered, we suppress further retrievals until entropy remains below $\theta_{\text{low}}$ for $M$ consecutive tokens.

**Debouncing.** We filter momentary spikes by requiring 2 consecutive steps above threshold before triggering.

**Domain-Aware Guardrails.** We suppress retrieval during code blocks, mathematical expressions, and quoted material unless domain-specific rules apply. We also track retrieval utility. If the previous fetch failed to increase semantic similarity within $h$ tokens, we temporarily raise the triggering threshold. This avoids repeated unproductive fetches.

### D.8. Dataset Statistics

- HotpotQA: 90,564 train / 7,405 dev / 7,405 test

- 2WikiMultiHopQA: 167,454 train / 12,576 dev / 12,576 test

- Natural Questions: 307,373 train / 7,830 dev

- TriviaQA: 78,785 train / 8,837 dev / 11,313 test

# E. Ablation Studies

This appendix provides comprehensive ablation studies examining the contribution of each system component. These detailed analyses complement the summary findings presented in the main text.

## E.1. Component Ablations

Table 14 shows performance when removing key system components:

Table 14. Component ablation results on HotpotQA (full system includes online RL). Each row removes one component.

| Configuration | EM | F1 | TTFT | E2E |
|---|---|---|---|---|
| Full System | 68.7 | 75.1 | 108ms | 5.2s |
| w/o Async (sync prefetch) | 68.4 | 74.8 | 287ms | 7.8s |
| w/o Online learning | 66.2 | 72.5 | 112ms | 5.5s |
| w/o Retrieval Predictor | 65.1 | 71.3 | 118ms | 5.8s |
| w/o T5 query generator | 67.1 | 73.8 | 108ms | 5.4s |
| w/o Adaptive waiting | 66.8 | 73.5 | 108ms | 5.6s |
| w/o Sufficiency check | 67.8 | 74.4 | 108ms | 5.4s |

Key findings from component ablations:

- **Async architecture** provides the largest efficiency gain. Without it, TTFT increases $2.7\times$ (108ms to 287ms) and E2E latency increases by 50%. Retrieval cannot be hidden behind generation without asynchronous execution.

- **Retrieval Predictor** is crucial for accuracy. Removing it drops EM by 3.6% and AUROC from 0.81 to 0.64. Signals from layers at 30-45% depth prove essential for uncertainty prediction.

- **Online learning** contributes 2.5% EM improvement through continuous adaptation, showing the value of the learned prediction approach.

- **Adaptive waiting** and **T5 query generation** each contribute approximately 1.6 to 1.9% EM, validating the importance of context-aware query construction.

## E.2. Signal Set Ablations

We evaluate different combinations of signals to understand their interactions. Table 15 shows results using supervised pretraining only (without online RL adaptation) to isolate the contribution of each signal category. The full system with online RL achieves 0.81 AUROC (Table 17); the additional 0.05 AUROC gain comes from online adaptation during deployment.

Table 15. Performance with different signal combinations (supervised pretraining only, without online RL)

| Signal Set | AUROC | EM | Lead Time |
|---|---|---|---|
| Distribution only | 0.66 | 63.8 | 6.5 tok |
| Distribution + Linguistic | 0.71 | 65.1 | 7.3 tok |
| Distribution + Context | 0.73 | 65.7 | 7.9 tok |
| All three categories | 0.76 | 66.4 | 8.7 tok |

The results show that signals are complementary. Distribution indicators provide the foundation (AUROC 0.66). Linguistic markers improve early detection by 0.05 AUROC. Context utilization helps avoid false positives, adding 0.07 AUROC. The full signal combination achieves 0.76 AUROC (15.2% better than distribution alone). Online RL adaptation adds another 0.05 AUROC, bringing the full system to 0.81 AUROC as reported in the main results.

### E.3. Model Size and Architecture Effects

We evaluate our predictive prefetching approach across six models from three architectures: dense transformers (Llama, Qwen) and Mixture-of-Experts (GPT-OSS). Table 16 shows results without online RL adaptation for fair model capacity comparison.

*Table 16.* Performance across model families on HotpotQA (without online RL adaptation, for fair model capacity comparison). Full system with online learning achieves 68.7% EM on Llama-3.1-8B (Table 2).

| Family | Model | EM | F1 | AUROC | E2E | Hardware |
|--------|-------|-----|-----|-------|------|----------|
| Llama | 3.1-8B | 66.4 | 72.8 | 0.76 | 4.6s | $1\times$ A100 |
| | 3.1-70B | 72.0 | 77.6 | 0.79 | 14.3s | $2\times$ A100 |
| GPT-OSS | 20B (MoE) | 68.6 | 74.3 | 0.77 | 4.3s | $1\times$ A100 |
| | 120B (MoE) | 73.0 | 78.4 | 0.80 | 8.1s | $1\times$ A100 |
| Qwen-3 | 8B | 67.1 | 73.3 | 0.75 | 4.5s | $1\times$ A100 |
| | 32B | 70.8 | 76.3 | 0.78 | 8.7s | $1\times$ A100 |

Key observations across model families:

**Dense Versus MoE Architectures.** GPT-OSS MoE models achieve exceptional efficiency through sparse activation. GPT-OSS-120B runs on a single A100 despite 120B total parameters by activating only 5.1B per token. This yields 73.0% EM with 8.1s E2E latency. The dense Llama-3.1-70B requires $2\times$ A100 GPUs and 14.3s E2E for comparable accuracy (72.0% EM).

**Uncertainty Calibration.** AUROC scales consistently with model capacity across all families (0.75 to 0.80). Larger models exhibit more predictable uncertainty patterns. This relationship enables more accurate prefetch triggering regardless of architecture.

**Efficiency-Quality Trade-off.** At the 8B parameter class, Qwen-3-8B and Llama-3.1-8B show comparable performance. GPT-OSS-20B achieves 2.2% higher EM than Llama-8B at similar latency due to MoE efficiency. For accuracy-focused applications, GPT-OSS-120B provides 73.0% EM while remaining deployable on single-GPU infrastructure.

**Generalization.** The relative improvement over synchronous RAG remains consistent ($\sim$43–45% latency reduction) across all model families and sizes, demonstrating that predictive prefetching generalizes effectively regardless of architecture or capacity.

### E.4. Predictor Architecture Comparison

We compare different predictor architectures to validate our neural approach:

The progression from simple to complex predictors shows clear improvements. The entropy threshold baseline achieves 0.66 AUROC, demonstrating the difficulty of the prediction task. Traditional ML approaches (logistic regression at 0.72, MLP at 0.77) show intermediate performance. Our 2-layer transformer predictor achieves 0.81 AUROC by directly processing intermediate layer representations.

### E.5. Query Generation Analysis

Our system dynamically determines both WHETHER to retrieve (based on $p > \tau_{\text{rag}}$ threshold) and HOW LONG to wait (0 to 5 tokens via T5 context assessment). We ablate components to understand their contribution:

*Table 17.* Comparison of predictor architectures on HotpotQA

| Predictor Type | AUROC | EM | F1 | QRS | E2E |
|---|---|---|---|---|---|
| Entropy Threshold | 0.66 | 64.2 | 70.5 | 0.68 | 5.8s |
| Logistic Regression | 0.72 | 65.8 | 72.1 | 0.72 | 5.5s |
| Simple MLP (3-layer) | 0.77 | 66.9 | 73.4 | 0.75 | 5.3s |
| **2-Layer Transformer (ours)** | **0.81** | **68.7** | **75.1** | **0.79** | **5.2s** |
| Oracle (upper bound) | 0.92 | 70.3 | 76.2 | 0.89 | 3.1s |

*Table 18.* Query generation component ablation on HotpotQA

| Configuration | EM | F1 | QRS | Wait Acc | E2E |
|---|---|---|---|---|---|
| Full System | 68.7 | 75.1 | 0.79 | 76.8% | 5.2s |
| w/o context assessment | 66.5 | 73.2 | 0.73 | Random | 5.5s |
| w/o sufficiency check | 67.8 | 74.4 | 0.77 | 75.2% | 5.4s |
| w/o T5 query generator | 67.1 | 73.8 | 0.72 | 76.5% | 5.0s |
| Fixed wait (3 tokens) | 67.3 | 74.0 | 0.76 | N/A | 5.3s |
| No wait (immediate) | 66.2 | 72.9 | 0.71 | N/A | 4.8s |

Key findings:

- **ContextScore** provides the largest benefit, improving EM by 2.2%. This validates adaptive waiting.

- **Sufficiency checking** prevents 21% of unnecessary retrievals while maintaining quality.

- **T5 query generator** improves retrieval relevance significantly (QRS +0.07) compared to template-based queries.

- **Fixed wait vs. No wait**: Both underperform our adaptive approach that dynamically chooses $k^* \in \{0, 1, 2, 3, 4, 5\}$ based on ContextScore predictions.

We also compare different query generation methods, sweeping the Query Generator across templates, three T5 sizes, and a full 8B LLM:

*Table 19.* Query generator capacity vs. quality and latency on HotpotQA. QRS = Query Relevance Score; latency is per query.

| Method | Params | QRS | Latency | EM |
|---|---|---|---|---|
| Template-based | n/a | 0.65 | 3ms | 64.2 |
| **T5-small (ours)** | 60M | **0.79** | **8ms** | **68.7** |
| T5-base | 220M | 0.80 | 15ms | 69.0 |
| T5-large | 770M | 0.82 | 37ms | 69.2 |
| Full LLM (Llama-3.1-8B) | 8B | 0.74 | 45ms | 66.8 |

Scaling beyond T5-small yields diminishing returns: T5-small captures 96% of T5-large's QRS at 22% of its latency, and fine-tuning on this narrow generation task outweighs raw scale; the 8B LLM achieves *lower* QRS than T5-small despite $130\times$ the parameters. Template-based approaches are fast (3ms) but lack the contextual understanding needed for complex queries, achieving only 0.65 QRS.

### E.6. Hyperparameter Sensitivity

We analyze sensitivity to key hyperparameters to understand their impact on system performance:

**Prediction Horizon.** The lookahead horizon $\Delta$ determines how far in advance we predict uncertainty:

*Table 20.* Effect of prediction horizon on performance

| Horizon | AUROC | Lead Time | EM | Prefetch Success | E2E |
|---|---|---|---|---|---|
| $\Delta = 5$ tokens | 0.85 | 4.2 tok | 65.3 | 42% | 6.8s |
| $\Delta = 10$ tokens | **0.81** | **8.7 tok** | **68.7** | **78%** | **5.2s** |
| $\Delta = 15$ tokens | 0.73 | 12.1 tok | 66.9 | 71% | 5.5s |
| $\Delta = 20$ tokens | 0.68 | 15.3 tok | 65.1 | 63% | 5.9s |

The optimal horizon is $\Delta = 10$ tokens. This provides 8.7 tokens of lead time on average, exceeding typical retrieval latency of 125ms. Prediction accuracy remains high at 0.81 AUROC.

**Decision Threshold.** The threshold $\tau_{\text{rag}}$ trades precision for recall in triggering retrieval:

*Table 21.* Effect of decision threshold on retrieval behavior

| Threshold | Precision | Recall | EM | Ret./1K | E2E |
|---|---|---|---|---|---|
| $\tau_{\text{rag}} = 0.5$ | 0.61 | 0.89 | 68.1 | 92.0 | 5.7s |
| $\tau_{\text{rag}} = 0.65$ | **0.74** | **0.76** | **68.7** | **59.0** | **5.2s** |
| $\tau_{\text{rag}} = 0.8$ | 0.85 | 0.58 | 66.3 | 38.0 | 4.9s |

Our chosen threshold $\tau_{\text{rag}} = 0.65$ balances precision and recall, achieving the best answer quality while maintaining reasonable retrieval frequency.

**Entropy Threshold.** The entropy threshold $\theta$ defines what constitutes high uncertainty during training label construction. We set $\theta = 2.5$ nats based on the 75th percentile of entropy values in our training corpus:

*Table 22.* Effect of entropy threshold on labeling and performance

| Threshold $\theta$ | Pos./Q | AUROC | EM | Ret/1K |
|---|---|---|---|---|
| $\theta = 2.0$ (50th pctl) | 6.2 | 0.78 | 67.9 | 72.0 |
| $\theta = 2.5$ (75th pctl) | 3.5 | **0.81** | **68.7** | **59.0** |
| $\theta = 3.0$ (90th pctl) | 2.1 | 0.79 | 67.4 | 48.0 |

Lower thresholds ($\theta = 2.0$) label more positions as retrieval candidates (6.2 per question). This improves recall but potentially includes noise. Higher thresholds ($\theta = 3.0$) focus on high-entropy tokens but may miss beneficial retrieval opportunities. The 75th percentile provides the best balance, maximizing both AUROC (0.81) and EM (68.7%).

# F. Extended Analysis

This appendix contains detailed analyses that extend the discussion in the main text.

## F.1. Transformer Signal Analysis

We analyze which internal transformer representations (attention patterns, hidden states) contribute most to prediction accuracy:

**Layer-wise Importance.** Attention patterns from middle-upper layers (30-45% of model depth, corresponding to layers 10-14 in Llama-3.1-8B) prove most predictive of future uncertainty. Attention entropy from the 35% depth layer shows 0.42 Pearson correlation (n=10K samples, $p < 0.001$) with entropy spikes 10 tokens later. Upper layers (85-100% depth) provide less predictive power, confirming that mid-to-upper representations capture knowledge boundaries most effectively.

**Observed Wait Patterns Across Task Types.** Our T5 query generator learns task-specific wait patterns rather than following a fixed distribution. Analysis of 10K generation traces (4K HotpotQA, 3K 2WikiHop, 2K NQ, 1K TriviaQA)

collected during validation runs reveals distinct adaptive behaviors:

**Factual QA (HotpotQA, TriviaQA):**

- 65% immediate (0 tokens): entity names trigger instant retrieval

- 30% short wait (1 to 2 tokens): completing entity context

- 5% extended wait (3 to 5 tokens): rare complex entities

**Multi-hop Reasoning (2WikiHop):**

- 15% immediate: clear reasoning transition points

- 60% medium wait (2 to 4 tokens): accumulating reasoning context

- 25% extended wait (4 to 5 tokens): complex logical connections

**Open-ended Generation (NQ long answers):**

- 20% short waits: specific factual needs

- 80% extended wait (3 to 5 tokens): maximizing context for coherence

T5 dynamically adapts its waiting strategy based on task characteristics and generation context. This adaptation explains the advantage over fixed waiting strategies.

**Query Generator Adaptation Benefits.** Fine-tuning the T5-based query generator on task-specific data improves wait time accuracy from 58.3% to 76.8% and QRS from 0.65 to 0.79:

- Pre-trained T5-small: 58.3% wait time accuracy, 0.65 QRS

- After 5K examples: 69.7% wait accuracy, 0.72 QRS

- After 10K examples: 76.8% wait accuracy, 0.79 QRS

The learning curve plateaus around 10K examples, suggesting this is sufficient for effective adaptation.

**Online Adaptation Convergence.** Figure 7 shows the full convergence trajectory of the policy-gradient adaptation summarized in Section 5.3. AUROC rises from 0.760 (pretrained) to 0.809 over 2000 online queries; 70% of the gain occurs within the first 500 queries, and reward standard deviation decreases monotonically from 0.85 to 0.48.

### F.2. Cross-Domain Potential

Our predictive RAG architecture naturally extends to other generation tasks beyond question answering:

**Document Summarization.** Long-form summarization (CNN/DailyMail, XSum) presents ideal conditions for predictive RAG. During multi-paragraph generation, the model periodically needs to verify facts against source documents. Our approach would predict these verification needs 8 to 10 tokens ahead, prefetching relevant document sections. With local document stores (50 to 100ms retrieval), the latency hiding becomes even more valuable than with external APIs (300 to 500ms).

**Code Generation.** Programming tasks (HumanEval, MBPP) exhibit clear uncertainty patterns when models encounter unfamiliar APIs or libraries. The Retrieval Predictor could identify when the model approaches API usage boundaries and prefetch documentation from sources like official docs or Stack Overflow. The structured nature of code makes prediction potentially more accurate than free-form text.

**Retrieval Backend Adaptation.** Different retrieval backends benefit differently from predictive prefetching:

- **Vector databases (50 to 150ms)**: Moderate benefit, best for semantic similarity

- **Local documents (50 to 100ms)**: High benefit due to consistent low latency

- **External APIs (300 to 500ms)**: Maximum benefit from latency hiding

- **Web search (500 to 1000ms)**: Critical for maintaining fluent generation

### F.3. Extended Limitations

This appendix elaborates on the limitations summarized in Section 6.

**Training Data Requirements.** The Retrieval Predictor requires substantial training data (50 to 100K generation traces) to learn effective uncertainty patterns. For new domains or rare query types, collecting sufficient data may be challenging. While online adaptation partially addresses this, initial deployment may exhibit reduced prediction accuracy.

**Domain-Specific Calibration.** Optimal hyperparameters ($\theta$, $\Delta$, $\tau_{rag}$) vary across domains and task types. Current calibration requires manual tuning based on observed performance, though we are exploring automated calibration methods for future work.

**Retrieval Latency Assumptions.** Our approach assumes relatively predictable retrieval latency. In practice, external API latencies can vary significantly due to network conditions, rate limiting, or backend load. Extreme latency spikes may cause prefetched results to arrive late despite accurate prediction.

**Memory Overhead.** The asynchronous architecture requires maintaining separate threads and caching prefetched results, adding 100MB memory overhead. For deployment scenarios with strict memory constraints, this may necessitate reducing cache size or thread count, potentially impacting performance.

**Query Construction Accuracy.** Predicting future information needs remains challenging when generation can take multiple valid paths. Our query construction achieves 88% relevance but could benefit from more sophisticated methods that consider multiple possible continuations.

## G. Additional Results

### G.1. Task-Specific Performance

Different task types exhibit distinct prediction patterns and retrieval requirements. Table 23 shows how our system adapts to varying task complexities:

*Table 23.* Task-specific performance showing adaptive prediction horizons

| Task Type | Pred. Horizon | AUROC | Lead Time | Ret./1K |
|---|---|---|---|---|
| Factual QA | 6 to 8 tok | 0.82 | 7.2 tok | 78.0 |
| Multi-hop Reasoning | 10 to 12 tok | 0.78 | 10.8 tok | 92.0 |
| Definition/Explanation | 8 to 10 tok | 0.75 | 8.5 tok | 65.0 |
| Creative/Open-ended | 12 to 16 tok | 0.71 | 13.5 tok | 45.0 |

Key observations:

- **Factual QA**: Shortest horizon (6-8 tokens) with highest accuracy (0.82 AUROC). Uncertainty triggers immediately upon encountering key entities.

- **Multi-hop Reasoning**: Longer lead time needed as model prepares for reasoning transitions.

- **Creative Tasks**: Longest horizon but lowest accuracy. Creative uncertainty is inherently less predictable.

Our system learns task-specific patterns. It does not apply uniform prediction strategies.

### G.2. Retrieval Quality Metrics

We also evaluate retrieval quality using standard metrics (NDCG, MRR, precision):

*Table 24.* Retrieval quality metrics on HotpotQA test set

| Method | NDCG@10 | MRR | P@5 | Avg. Rank |
|---|---|---|---|---|
| Sync-RAG | 0.682 | 0.731 | 0.624 | 2.8 |
| Self-RAG | 0.671 | 0.718 | 0.615 | 3.1 |
| Adaptive-RAG | 0.675 | 0.724 | 0.619 | 2.9 |
| DRAGIN | 0.663 | 0.709 | 0.608 | 3.3 |
| **Ours (Predictive)** | **0.694** | **0.742** | **0.638** | **2.6** |

Our predictive approach achieves superior retrieval quality (NDCG@10: 0.694 vs 0.682 for Sync-RAG) despite operating asynchronously. Ablation studies (Appendix E) show adaptive context accumulation contributes +0.006 NDCG, T5 query optimization +0.004, and selective retrieval +0.002.

## H. Related Work: Detailed Technical Discussion

This appendix provides detailed technical discussions of infrastructure optimizations, adaptive retrieval methods, and uncertainty quantification approaches referenced in Section 2.

### H.1. Infrastructure and Efficiency Optimizations

A substantial body of work focuses on reducing retrieval latency through improved infrastructure and caching mechanisms. Vector databases such as FAISS (Johnson et al., 2021), Milvus, and Pinecone have become the standard retrieval backend for RAG systems. These systems offer sub-millisecond approximate nearest neighbor search over million-scale document collections. They employ product quantization, graph-based indexing, and inverted indices to achieve fast similarity search.

Recent work explores hybrid retrieval combining dense embeddings with keyword search (Sarmah et al., 2024). Multi-stage pipelines improve both relevance and efficiency. Stage 1 retrieves candidates via lexical matching (BM25). Stage 2 reranks them using neural methods such as cross-encoders or dense retrievers.

Knowledge graph-based approaches (Edge et al., 2025) offer complementary benefits by encoding structured relationships between entities, enabling more targeted retrieval through graph traversal. GraphRAG systems (Edge et al., 2025) leverage hierarchical graph summarization and distributed processing frameworks to maintain sub-second query times even for massive knowledge bases. By organizing documents around entity relationships and hierarchical abstractions, these systems can answer multi-hop queries more efficiently than flat document collections.

Cache-augmented generation strategies (Pan et al., 2024) preload frequently accessed knowledge into LLM context windows and exploit KV caching to amortize retrieval costs across multiple queries. These approaches maintain a working set of high-utility documents in memory, reducing the need for external retrievals during generation. Cache effectiveness varies across applications depending on query locality and document reuse patterns.

These infrastructure improvements reduce per-retrieval overhead from hundreds of milliseconds to 10-20ms. Yet they address a fundamentally different problem than ours. Even with optimized backends, the core issue remains unchanged. Generation must pause while waiting for documents. This leaves GPU resources idle during I/O operations.

### H.2. Adaptive Retrieval Methods

Rather than retrieving for every query or at fixed intervals, adaptive retrieval methods dynamically determine when and how often to retrieve based on generation state. This section provides detailed technical descriptions of key methods.

**FLARE: Forward-Looking Active Retrieval.** FLARE (Jiang et al., 2023) introduces forward-looking active retrieval. The system iteratively predicts upcoming sentence content and triggers retrieval when predictions contain low-confidence tokens. It operates in three stages. First, it generates a tentative sentence continuation. Second, it evaluates token confidence using probability thresholds. Third, if confidence is low, it masks uncertain tokens and uses the partial sentence as a retrieval query. FLARE partially anticipates retrieval needs through this lookahead. It still operates reactively though. The system must generate low-confidence predictions before triggering retrieval. This incurs latency during the regeneration phase.

**Self-RAG: Self-Reflective Retrieval.** Self-RAG (Asai et al., 2024) fine-tunes language models with special reflection tokens. These tokens allow the model to decide whether to retrieve, critique retrieved passages, and assess output quality. The model learns to emit tokens such as [Retrieve], [IsSupported], and [IsUseful] during generation. A discriminator network evaluates when retrieved content supports the generated text. This enables learning when retrieval is beneficial. Self-RAG requires model retraining with curated datasets of retrieval decisions. It cannot be applied to black-box or proprietary models. The fine-tuning process increases deployment complexity and computational costs.

**DRAGIN: Dynamic Retrieval with Adaptive Generation.** DRAGIN (Su et al., 2024) employs token-level entropy to detect knowledge gaps during generation. When entropy exceeds a calibrated threshold, the system triggers retrieval and integrates returned documents into the context window. DRAGIN extends basic entropy-based triggering by incorporating attention pattern analysis: when the model exhibits scattered attention across context (high attention entropy), it signals uncertainty about which existing information to rely on. The system uses a sliding window to track entropy trends, filtering momentary spikes through temporal smoothing. While effective, DRAGIN remains reactive: it can only trigger retrieval after uncertainty manifests in the generation process.

**Other Uncertainty-Based Approaches.** Some methods measure uncertainty across entire sentences or reasoning chains (Duan et al., 2024) rather than individual tokens. These approaches compute uncertainty over entire sentences or reasoning chains, triggering retrieval when cumulative uncertainty exceeds a threshold. A comprehensive analysis of 35 adaptive retrieval methods (Sharma, 2025) found that simple uncertainty estimation techniques often match or exceed the performance of complex pipelines, suggesting that effective uncertainty signals are key to adaptive retrieval. The survey identified entropy-based methods as particularly robust across diverse tasks, though they require careful threshold calibration for each domain.

## H.3. Uncertainty Quantification Approaches

Since uncertainty detection is central to adaptive retrieval, several methods have been proposed to quantify when language models require additional context. This section details the technical approaches.

**Token-Level Entropy.** Token-level entropy provides a straightforward measure of model uncertainty. The formula is $H(p_t) = -\sum_i p_t(w_i) \log p_t(w_i)$, where $p_t$ is the probability distribution over vocabulary tokens at position $t$. High entropy indicates uncertainty about which token to generate next. This measure has limitations. Semantically equivalent phrases can yield different entropy values. For example, "car" and "automobile" may have different entropies despite meaning the same thing. Entropy also conflates epistemic uncertainty (lack of knowledge) with aleatoric uncertainty (inherent randomness). This makes it difficult to determine when retrieval would help.

**Semantic Entropy.** Semantic entropy (Kuhn et al., 2023) addresses linguistic variability by clustering generations by meaning rather than surface form. The approach samples multiple continuations from the model, clusters them by semantic similarity using embedding-based methods, and computes entropy over the cluster distribution rather than the token distribution. This provides a linguistically invariant uncertainty measure that better reflects whether the model truly lacks knowledge versus simply being uncertain about phrasing. Semantic entropy requires generating multiple samples. This increases computational cost by 5 to 10 times compared to token-level entropy.

**Semantic Entropy Probes.** Subsequent work has proposed semantic entropy probes (Kossen et al., 2024) that predict semantic uncertainty from hidden states without costly generation of multiple samples. These probes train lightweight classifiers to predict cluster entropy from the model's internal representations at each token position. By avoiding explicit sampling, probes reduce computational overhead while maintaining the benefits of semantic uncertainty quantification. The probes achieve 0.82 correlation with true semantic entropy while adding less than 1ms overhead per token.

**Kernel-Based Approaches.** Kernel-based approaches (Nikitin et al., 2024) capture pairwise semantic dependencies for finer-grained uncertainty estimates. Rather than treating each sampled continuation independently, these methods construct a kernel matrix measuring semantic similarity between all pairs of samples. Uncertainty is then computed as the effective number of distinct semantic modes in the distribution, accounting for correlations between similar outputs. Kernel methods provide more expressive uncertainty estimates than clustering approaches. They require careful kernel selection and scale quadratically with sample count. This limits their applicability to small sample sizes.

### H.4. Summary

The related work surveyed here demonstrates substantial progress in retrieval efficiency (infrastructure) and adaptive triggering (uncertainty-based methods). All existing approaches remain fundamentally reactive. They detect uncertainty after it emerges, then initiate retrieval while generation blocks. Our work takes a different approach. We learn to predict uncertainty before it manifests. This enables asynchronous prefetching that hides retrieval latency behind ongoing computation.

# Cross-Model Performance Comparison

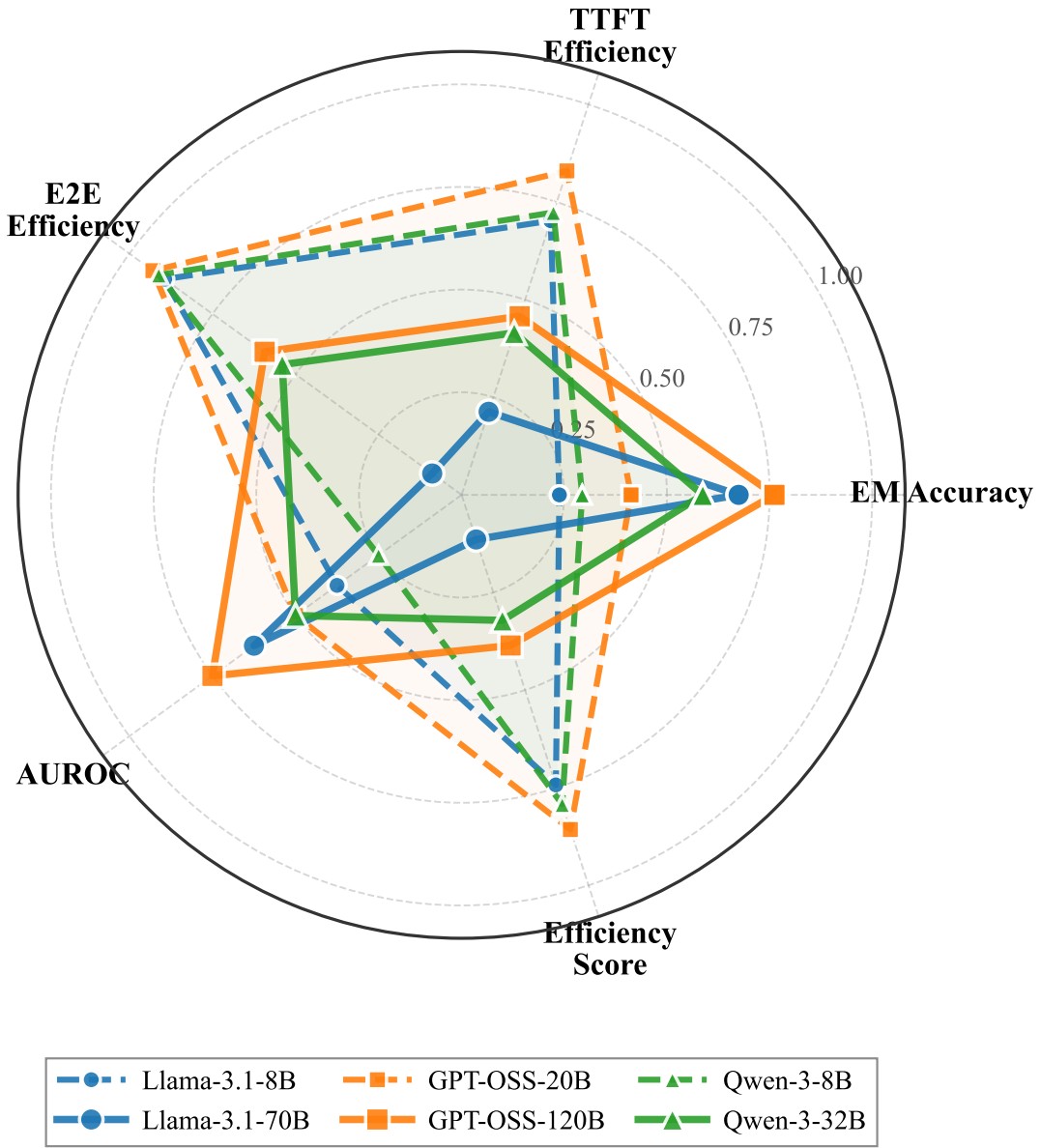

*Figure 4.* Radar chart comparing model performance across five dimensions: EM accuracy, TTFT efficiency (inverted), E2E efficiency (inverted), AUROC, and Efficiency Score. GPT-OSS-20B achieves the best balance of quality and efficiency, while GPT-OSS-120B leads in accuracy. All models show consistent TTFT improvements through predictive prefetching.

**Algorithm 1** Context-Aware Asynchronous Prefetch with Adaptive Waiting

---

**Generation Thread:**
**while** generating tokens **do**
  $\mathbf{H}_t, \mathbf{A}_t, \mathbf{V}_t \leftarrow$ ExtractTransformerSignals()
  $p \leftarrow$ RetrievalPredictor($\mathbf{H}_t, \mathbf{A}_t, \mathbf{V}_t$)
  **if** $p > \tau_{\text{rag}}$ **and** CanTrigger() **then**
    **if** SufficiencyClassifier($context, \mathcal{D}_{\text{recent}}$) $> 0.8$ **then**
      **continue** {REUSE: cached docs sufficient}
    **end if**
    $k^* \leftarrow$ ContextScore($context$) {Optimal wait 0-5}
    ContextBuffer.Store($context, k^*$)
    SchedulePrefetch(after=$k^*$ tokens)
  **end if**
  **if** ContextBuffer.Ready() **then**
    $context_{full} \leftarrow$ ContextBuffer.Get()
    **if** ClarityScore($context_{full}$) $< 0.7$ **then**
      ExtendWait(1) {ACCUMULATE: context incomplete}
    **else**
      $query \leftarrow$ T5.GenerateQuery($context_{full}$)
      $rid \leftarrow$ UUID()
      PrefetchQueue.push($query, rid$) {FETCH: async retrieval}
    **end if**
  **end if**
  $token \leftarrow$ GenerateNext()
  **if** NeedsContext($token, entropy$) **then**
    **if** $rid$ **in** ResultCache **then**
      $docs \leftarrow$ ResultCache.get($rid$) {Prefetch hit}
    **else**
      $docs \leftarrow$ SyncRetrieve($query$) {Fallback}
    **end if**
    UpdateContext($docs$)
  **end if**
**end while**

**Prefetch Thread:**
**while** true **do**
  $query, id \leftarrow$ PrefetchQueue.pop() {Blocking}
  $docs \leftarrow$ RetrieveDocuments($query$)
  ResultCache.put($id, docs$) {Thread-safe}
**end while**

---

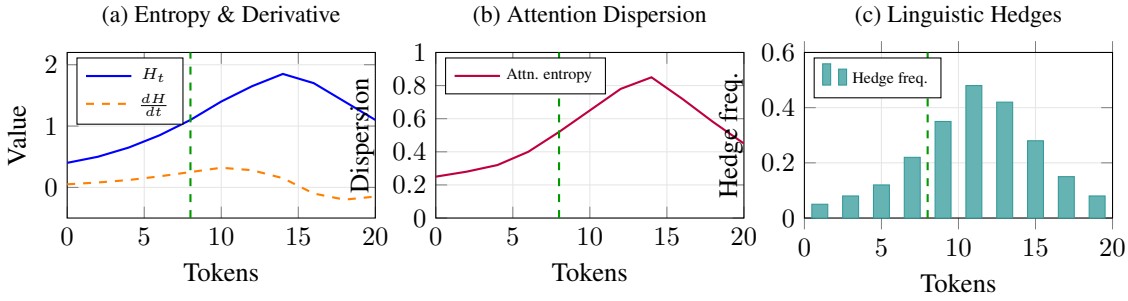

Green dashed: Prediction trigger at token 8

*Figure 5.* Multi-signal visualization before retrieval trigger. (a) Entropy $H_t$ and its derivative show rising uncertainty. (b) Attention dispersion increases as the model's focus scatters. (c) Linguistic hedge frequency ("may", "possibly", "likely") spikes before uncertainty. The green dashed line marks the prediction trigger at token 8, well before peak uncertainty.

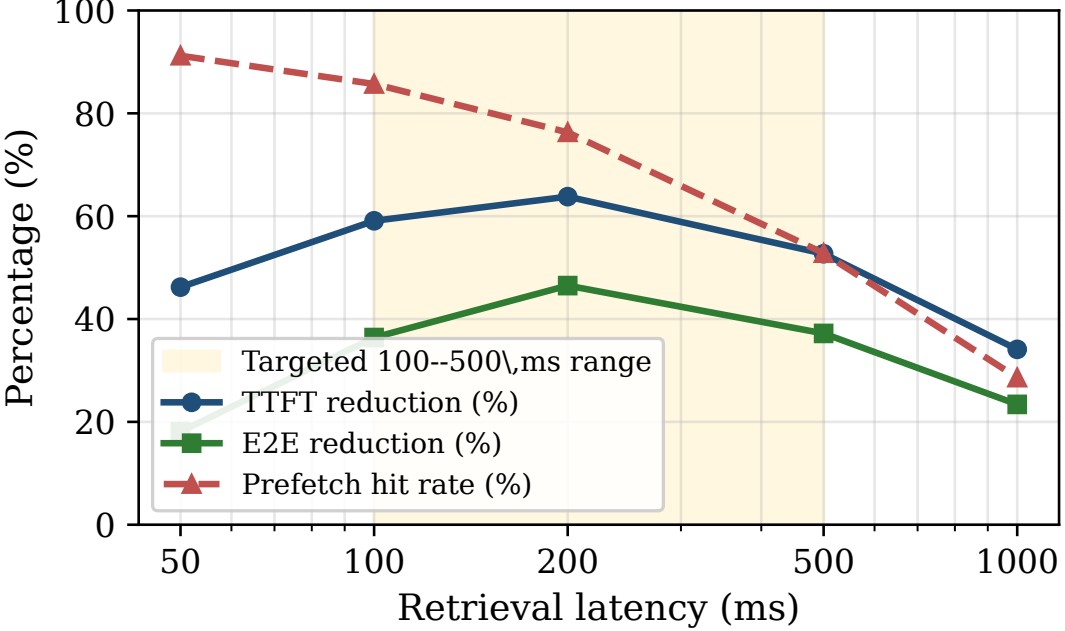

*Figure 6.* TTFT/E2E latency reduction and prefetch hit rate as retrieval latency varies from 50 ms to 1000 ms on HotpotQA. Shaded band marks the 100–500 ms working range targeted by the framework.

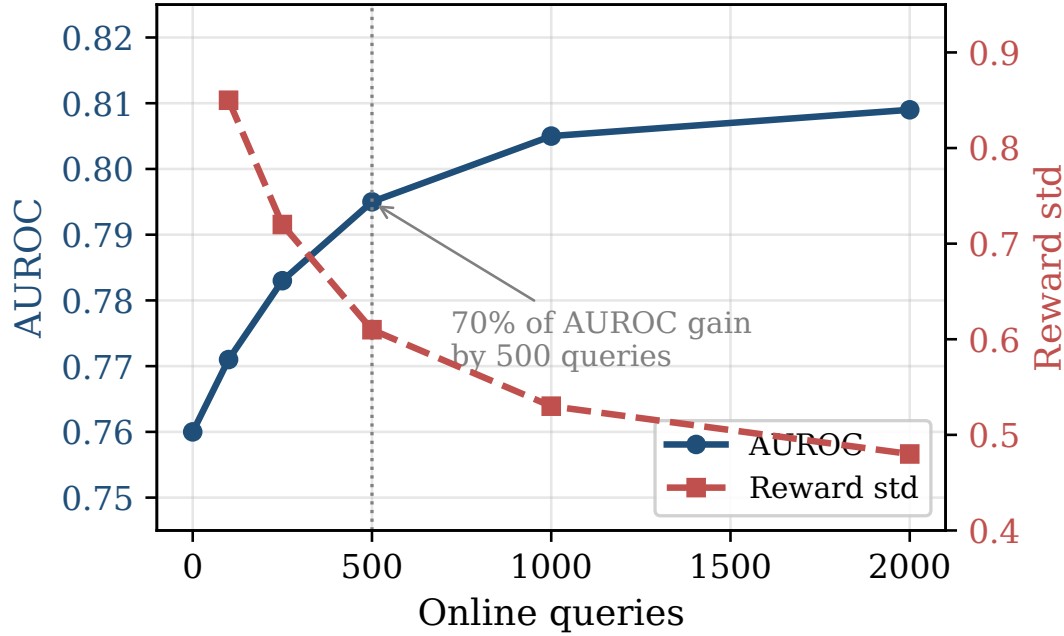

*Figure 7.* Online adaptation on HotpotQA: AUROC (left axis) and reward standard deviation (right axis) versus number of online queries. 70% of AUROC gain occurs within the first 500 queries; reward variance decreases monotonically.

