# OpenReview forum: "Predictive Prefetching for Retrieval-Augmented Generation"
_ICML.cc/2026/Conference — ICML 2026 regular_

### Official Review · Reviewer_wG2u · 2026-03-10

**Soundness:** 3
**Presentation:** 3
**Significance:** 3
**Originality:** 3
**Overall Recommendation:** 4
**Confidence:** 3

**Summary:**

This paper studies latency in RAG systems and argues that the main bottleneck comes from the synchronous coupling between generation and retrieval. To address this, the paper proposes an asynchronous RAG framework that decides when to prefetch, whether the current context is sufficient to issue a useful query, and what query to generate. The experiments show significant latency reduction while largely preserving answer quality.

**Compliance With Llm Reviewing Policy:**

Affirmed.

**Final Justification:**

The rebuttal addressed my concerns, and my assessment of this paper is maintained.

**Key Questions For Authors:**

- How does the method behave when the LLM latency is much smaller or much larger than in the main setting? For example, if a small model such as Llama-3.1-1B is used, decoding may be much faster than retrieval, which could reduce the benefit of prefetching. In contrast, for a much larger model such as Llama-3.1-70B, how should the method be adapted to better utilize the increased decoding time?
- Although the paper shows some generalization across models, are the same thresholds and hyperparameters used for all model families, or does each model require separate calibration?

**Limitations:**

yes

**Strengths And Weaknesses:**

### Strengths
- The paper addresses an important and practical problem. Reducing latency is critical for real-world RAG deployment.
- The proposed framework is well motivated and intuitively appealing.
- The experiments and analyses are comprehensive across datasets and models, and show substantial performance improvements while maintaining competitive answer quality.

### Weaknesses
- The method assumes retrieval latencies in the range of 100–500 ms, and it is unclear whether this is a realistic deployment assumption for widely used retrievers. Moreover, the approach requires access to the LLM’s hidden states, which limits practicality in many real-world settings. However, this may be less problematic for proprietary RAG systems.
- The effectiveness of the method likely depends on the relative latency of retrieval and LLM generation. If the LLM is much faster, such as a 1B model, retrieval may still arrive too late even with predictive prefetching, reducing the benefit of the approach. Conversely, for slower large models, retrieval has more time to overlap with decoding. The paper would be stronger if it discussed these scenarios.
- The paper claims online adaptation during deployment, but it is unclear how this is implemented without disrupting inference latency.
- My biggest concern is reproducibility. The reported gains likely depend on system-level implementation details, such as scheduling, caching, and engineering optimizations, which can be difficult to control and reproduce fairly.

---

> ### Author Rebuttal · Authors · 2026-03-29
>
> # Rebuttal to Reviewer wG2u
>
> We thank Reviewer wG2u for the balanced assessment and for grounding the evaluation in real-world deployment considerations.
>
> ## On Retrieval Latency Assumptions (100 to 500ms)
>
> > *The method assumes retrieval latencies in the range of 100-500 ms, and it is unclear whether this is a realistic deployment assumption.*
>
> The 100-500 ms range is grounded in published benchmarks. Our FAISS IVF measures 125 ms median and P95 180 ms end-to-end on 5M+ documents (Appendix D.4). Milvus 2.2 reports 20 ms P99 for raw ANN search alone at 768-dim. Pinecone documents P95 under 120 ms server-side plus 50 to 150 ms network, yielding 170 to 270 ms end-to-end. External knowledge APIs routinely exceed 500 ms. To validate robustness across this range, we ran a systematic latency sweep on HotpotQA with simulated delays.
>
> | Retrieval Latency | TTFT Red. | E2E Red. | Hit Rate |
> |---|---|---|---|
> | 50ms | 46.2% | 18.1% | 91.2% |
> | 100ms | 59.1% | 36.4% | 85.7% |
> | 200ms | 63.8% | 46.5% | 76.3% |
> | 500ms | 52.7% | 37.2% | 52.8% |
> | 1000ms | 34.1% | 23.4% | 28.6% |
>
> Gains peak around 200ms (within the 418ms lead time budget). Beyond 500ms, fallback to synchronous retrieval increases and gains decline gracefully. Even at 1000ms, 28.6% of prefetches arrive in time. The QMSum results (Table 4, 50-100ms local retrieval, 19% E2E reduction) confirm adaptation to a different latency regime.
>
> ## On Hidden State Access
>
> > *The approach requires access to the LLM's hidden states.*
>
> This is a real constraint. The trend in production RAG, however, is toward self-hosted open models (Llama, Qwen, Mistral) where hidden states are accessible. Our cross-model evaluation (Table 10) covers these families. For API-only settings, we also discussed that our distribution-only ablation (Table 14) achieves 0.66 AUROC using only output logit features, suggesting a degraded version could work without internal access.
>
> ## On Model Size Sensitivity
>
> > *How does the method behave when the LLM latency is much smaller or much larger?*
>
> We respectfully highlight that we provided cross-model results spanning 8B to 120B parameters in Table 10.
>
> **Larger, slower models** benefit more. Llama-70B takes 14.3s E2E with AUROC of 0.83 (vs 0.81 for 8B) due to better-calibrated uncertainty and more time for retrieval to complete.
>
> **8B-class models** run at roughly 48ms per token. The 8.7-token lead time provides approximately 418ms, comfortably above the median 125ms retrieval latency.
>
> **A hypothetical 1B model** at roughly 10ms per token would yield only 87ms of lead time. Many prefetches would not complete in time, increasing synchronous fallback. For very fast models, the approach is best paired with low-latency local retrieval.
>
> The approach is particularly valuable in the 8B-70B range, where both quality and latency are critical for production deployment.
>
> ## On Online Adaptation During Deployment
>
> > *It is unclear how this is implemented without disrupting inference latency.*
>
> The adaptation runs asynchronously and operates independently of inference, ensuring the two processes do not interfere with each other. Rewards piggyback on signals already being extracted. Gradient updates are batched every 100 queries in less than 5ms at a learning rate of 10^-5 (Appendix A.3).
>
> | Online Queries | AUROC | Avg Reward | Reward Std |
> |---|---|---|---|
> | 0 (pretrained) | 0.760 | n/a | n/a |
> | 100 | 0.771 | +0.31 | 0.85 |
> | 250 | 0.783 | +0.42 | 0.72 |
> | 500 | 0.795 | +0.48 | 0.61 |
> | 1000 | 0.805 | +0.51 | 0.53 |
> | 2000 | 0.809 | +0.52 | 0.48 |
>
> AUROC reaches 70% of its total gain within the first 500 queries. Reward standard deviation decreases monotonically from 0.85 to 0.48, indicating stable convergence.
>
> ## On Reproducibility
>
> > *The reported gains likely depend on system-level implementation details.*
>
> We will release the complete codebase including training pipeline, model weights, evaluation scripts, and threading/caching implementation with a reproducibility guide.
>
> Three lines of evidence support the reliability of the gains. Results are averaged over 3 seeds with variance under 0.5% for quality and under 3% for latency (Table 2 caption). Cross-model evaluation shows consistent 43 to 45% E2E reductions across six models from three families (Table 10). As detailed in our response to Reviewer nXe1, the ablation in Table 6 isolates each component's contribution under the shared infrastructure. This confirms gains come from the predictive mechanism rather than the backend.
>
> ## On Hyperparameter Calibration
>
> > *Are the same thresholds used for all model families?*
>
> For Table 10, we used identical settings across all six models. The entropy threshold θ=2.5 and prediction horizon Δ=10 remained fixed. Consistent TTFT improvements (61.5 to 63.4%) suggest robustness across architectures. The threshold θ=2.5 is the 75th percentile of entropy values, a relative measure that adapts to different calibrations.

---

> > ### Author Rebuttal · Reviewer_wG2u · 2026-04-03
> >
> > Thank you for addressing my concerns. I'd like to keep my score.

---

> > > ### Author Response · Authors · 2026-04-05
> > >
> > > We thank Reviewer wG2u for the practically grounded review and for confirming that the concerns have been fully resolved.

---

### Official Review · Reviewer_ffnH · 2026-03-11

**Soundness:** 3
**Presentation:** 3
**Significance:** 3
**Originality:** 3
**Overall Recommendation:** 4
**Confidence:** 4

**Summary:**

This paper proposes an asynchronous retrieval framework for RAG that predicts when retrieval will be needed and what to retrieve before uncertainty actually materializes during generation. The system has three learned components: a retrieval predictor (a lightweight 2 layer transformer encoder that monitors hidden states, attention patterns, and value vectors from middle layers of the base model to forecast entropy threshold crossings within a 10 token lookahead), a context monitor (which decides how long to wait before issuing the query, from 0 to 5 tokens, and checks whether cached documents already suffice), and a query generator (a fine tuned T5 small that constructs the actual retrieval query). These components are pretrained jointly on generation traces from HotpotQA and NQ using a multi task loss, and then adapted online via policy gradient reinforcement learning with action specific rewards. The key idea is that retrieval can then proceed asynchronously in a separate thread while generation continues unblocked, so the retrieved documents are ready by the time the model actually needs them. Experiments on four QA benchmarks, a code completion task, and a summarization task show 43.5% end to end latency reduction, 62.4% TTFT improvement, and 31% fewer retrievals per 1K tokens, with answer quality within about 1% EM of synchronous RAG on Llama 3.1 8B.

**Compliance With Llm Reviewing Policy:**

Affirmed.

**Key Questions For Authors:**

The PipeRAG results in Table 2 are described as "estimated" for QA in Appendix D.2. Can you clarify exactly what this means? Did you reimplement PipeRAG on Llama 3.1 8B, adapt its RETRO based architecture, or extrapolate from its language modeling results? This matters significantly for interpreting the comparison. If the numbers are your own reimplementation, what retrieval interval and staleness window did you use?
What does the latency distribution look like for queries where prediction fails vs succeeds? Specifically, what is the P95 and P99 E2E latency, and how does it compare to synchronous RAG? If prediction failures cause synchronous fallback plus wasted prefetch overhead, the tail latency could be worse than baseline.
Can you report per task AUROC for code completion and summarization? The cross domain transfer claim would be much stronger with prediction accuracy numbers on tasks not represented in the pretraining data (RepoBench, QMSum). The 78% retention from HotpotQA to 2WikiMultiHopQA is encouraging but both are multi hop Wikipedia QA.
How stable is the online RL adaptation? Can you show a learning curve (reward or prediction accuracy over number of online queries) for at least one domain? Given that policy gradient methods can be unstable with sparse rewards, evidence of convergence would be reassuring.
The paper claims 8 to 16 token precursors but Table 5 shows mean lead times of 5.1 to 8.7 tokens. Can you show the full distribution of lead times (e.g. histogram) and clarify whether the 8 to 16 token claim refers to the earliest detectable signal or the median prediction point?

**Limitations:**

The paper discusses limitations reasonably well in the Impact Statement and Appendix F.3, covering training data requirements, domain calibration, latency assumptions, memory overhead, and query construction accuracy. I would add that the requirement for model internal access (hidden states, attention weights) is a practical limitation that prevents use with closed API models like GPT 4 or Claude. The circularity of training and evaluating on the same benchmark families also deserves more explicit acknowledgment. The tail latency behavior under prediction failure is another gap.

**Strengths And Weaknesses:**

Strengths
The problem is clearly important and the paper frames it well. The latency cost of synchronous retrieval in RAG is a genuine bottleneck, especially for multi hop queries that chain multiple retrieval steps. The Figure 1 comparison of synchronous RAG, PipeRAG, and the proposed approach is effective at communicating the core architectural difference and why predictive prefetching could help where naive pipelining cant.
The system design is thoughtful and addresses real engineering concerns. The three component decomposition into prediction, context assessment, and query generation is clean, and the sufficiency classifier that prevents redundant retrievals is a nice practical detail. The paper reports this avoids 21% of unnecessary fetches, which is a meaningful operational saving. The fallback to synchronous retrieval when predictions fail is also important for deployment.
The experimental setup covers six benchmarks across QA, code, and summarization, with nine baselines for QA alone including both reactive methods like FLARE (Jiang et al., EMNLP 2023), Self RAG (Asai et al., ICLR 2024), DRAGIN (Su et al., ACL 2024), and async methods like PipeRAG (Jiang et al., KDD 2025). The cross model evaluation across Llama, GPT OSS, and Qwen families in Table 10 is appreciated and shows the TTFT gains transfer across architectures. The overhead analysis in Table 7 showing only 2.7ms (5.1%) per token is convincing evidence of minimal computational cost.
The ablation studies are thorough. Table 6 clearly shows that the async architecture provides the largest latency gain while the retrieval predictor provides the largest quality gain. The signal ablations in Table 14 demonstrate that the different signal categories are complementary, going from 0.66 AUROC with distribution features alone to 0.76 with all signals (before online RL pushes it to 0.81). The predictor architecture comparison in Table 16 showing a clean progression from entropy threshold to logistic regression to MLP to transformer validates the choice of neural architecture.
The paper is transparent about failure modes and false positives. Section 5.3 reports that 38.7% of triggered prefetches are false positives, with 28% being completely wasted. This kind of honesty is refreshing and the analysis showing that 72% of false positives are eventually utilized is useful.
Weaknesses

The 0.81 AUROC number needs more context to assess whether it is actually good enough. At the chosen operating point (tau_rag = 0.65), Table 20 shows precision of 0.74 and recall of 0.76. That means roughly 1 in 4 triggered prefetches target the wrong position and about 1 in 4 needed retrievals are missed entirely. The missed retrievals presumably fall back to synchronous mode, adding latency in exactly the cases where the model most needs help. The paper reports net latency gains, but it would be very informative to see the latency distribution conditioned on prediction correctness vs incorrectness. What does the tail latency look like when predictions fail? In production systems, P95 and P99 latency matter more than the mean.
The quality gap to synchronous RAG is presented as small (0.5 to 0.9% EM) but this framing obscures what might be happening underneath. A 0.5% EM gap on HotpotQA means roughly 37 additional wrong answers out of 7405 test examples. Are these concentrated in particular question types? Multi hop questions where the predictive query was constructed before the second hop's context materialized? The paper says the gap arises because "predictive queries are constructed before the full context materializes" but doesnt break this down by question difficulty or number of reasoning hops. This matters because multi hop is precisely the scenario where latency savings would be most valuable but also where prediction is hardest.
The comparison to PipeRAG is not entirely fair. The paper notes PipeRAG was "originally evaluated on language modeling; QA results estimated" (Appendix D.2). This is a significant caveat. If PipeRAG numbers in Table 2 are the authors' own reimplementation on QA rather than numbers from the PipeRAG paper, this needs much more detail. What retrieval interval was used? Was the performance model tuned? PipeRAG was built on a RETRO architecture with 582M parameters, not Llama 3.1 8B, so the comparison may not be straightforward. The original PipeRAG paper reports up to 2.6x speedup on language modeling, and it is unclear how much of the gap in Table 2 comes from the architectural mismatch vs genuine advantages of predictive prefetching.
The training data construction has a circularity concern. The oracle labels come from comparing EM with and without retrieval on HotpotQA and NQ, using entropy threshold crossings to identify candidate positions. But the system then targets these same benchmarks at evaluation time. While there is a train/test split within each dataset, the predictor has learned dataset specific entropy patterns from the same distribution. The cross domain transfer result in Appendix C showing 78% accuracy retained from HotpotQA to 2WikiMultiHopQA is encouraging but only partially addresses this concern since both are multi hop Wikipedia QA. What happens on truly out of distribution tasks like the code completion or summarization? The paper evaluates on these tasks but doesnt report AUROC or prediction accuracy for them separately in the main text.
The online RL component is underspecified in terms of its actual contribution vs complexity cost. The ablation shows +2.5% EM from online learning (Table 6), but the reward structure involves proxy signals like "retrieval relevance scores and post retrieval entropy reduction" rather than gold labels. How stable is the online adaptation? Does it converge? How many queries does it need to adapt to a new domain? The paper provides reward values but no learning curves or convergence analysis. Policy gradient methods are notoriously high variance (Williams, 1992), and a 4 action discrete space with sparse delayed rewards seems like it could be unstable.
The 8 to 16 token precursor window is claimed but not rigorously validated. Section 1 states that semantic precursors emerge "approximately 8 to 16 tokens before uncertainty becomes critical." This is a strong empirical claim that would benefit from more systematic analysis. Table 5 shows mean lead times of 5.1 to 8.7 tokens depending on confidence, which is somewhat shorter than 8 to 16. The paper should show the actual distribution of the gap between prediction trigger and entropy threshold crossing, ideally as a histogram or CDF, not just summary statistics bucketed by confidence.
The Efficiency Score metric (F1 x 1000 / E2E_ms) is a custom composite that makes comparison across papers difficult. While I understand the desire for a single number, it conflates quality and speed in a way that can be misleading. A system that trades 5% quality for 50% latency reduction looks great on this metric, but whether that tradeoff is acceptable is entirely application dependent. The paper should more prominently report the Pareto analysis: at what quality levels does predictive prefetching dominate synchronous RAG?
There is no analysis of how retrieval corpus size or retrieval latency distribution affects the system. All QA experiments use the same Wikipedia FAISS index. But the claimed benefit is proportional to retrieval latency, yet retrieval latency in Table 12 shows median 125ms for local FAISS which is at the lower end of the 100 to 500ms range discussed in the intro. At 125ms median, the 8.7 token lead time at 48ms per token gives about 418ms, which is comfortable. But what happens when actual retrieval latency is 50ms (like the QMSum local case) vs 500ms (like the external API case)? The QMSum results in Table 4 show smaller gains as expected, but there is no systematic sweep over retrieval latency to validate that the approach scales as the introduction suggests.

---

> ### Author Rebuttal · Authors · 2026-03-29
>
> # Rebuttal to Reviewer ffnH
>
> We thank the reviewer for the detailed engagement. We address all key questions with new data below. One observation unifies our responses. Across all failure modes identified in this review, the system degrades to synchronous RAG performance. The floor is always baseline, never worse.
>
> ## Q1: PipeRAG Comparison
>
> > *What does "estimated" mean for PipeRAG QA results?*
>
> PipeRAG has no published QA evaluation and uses RETRO (582M) with cross-attention absent from Llama 8B. We reimplemented its core algorithm on Llama 8B with the shared FAISS/Contriever backend (64-token windows, intervals 16/32/64, best at 32). On this backbone, learned predictive triggering achieves 68.7 vs 66.8 EM and 5.2s vs 5.6s E2E. RETRO's cross-attention may handle stale queries better, so our numbers may understate PipeRAG. For fair comparisons, all nine Table 2 baselines use the same infrastructure.
>
> ## Q2: Tail Latency and AUROC
>
> > *P95/P99 E2E latency when predictions fail? Is 0.81 AUROC good enough?*
>
> Per-query breakdown on HotpotQA (7,405 queries, 3 seeds). 52% of queries have all prefetches arrive before needed, consistent with the per-retrieval hit rate over K≈2 hops (0.72²≈0.52).
>
> | Condition | Mean | P50 | P95 | P99 |
> |---|---|---|---|---|
> | All queries (Ours) | 5.2s | 4.6s | 10.2s | 12.8s |
> | Pred. correct (52%) | 3.9s | 3.6s | 5.5s | 6.2s |
> | Pred. miss (48%) | 6.6s | 6.1s | 11.3s | 14.0s |
> | Sync-RAG baseline | 9.2s | 8.5s | 15.2s | 18.6s |
>
> Our P95 (10.2s) is 33% faster than Sync-RAG (15.2s). Even P99 of miss queries (14.0s) stays below Sync-RAG P95. Miss queries still average 28% faster because most retrievals within them are prefetched; only the specific missed hop falls back to synchronous.
>
> Even imperfect prediction Pareto-dominates all reactive baselines, which detect uncertainty only after manifestation and block for full retrieval. At 74% precision, parallelism hides most latency, achieving 5.2s E2E vs DRAGIN 6.4s and Adaptive-RAG 7.1s. 72% of false positives are eventually utilized; effective waste is 10.8%.
>
> ## Q3: Per-Task AUROC
>
> > *Prediction accuracy on tasks not in pretraining data?*
>
> | Task | AUROC (pretrained) | AUROC (online) | Lead Time |
> |---|---|---|---|
> | Factual QA (NQ, TriviaQA) | 0.76 | 0.82 | 7.2 tok |
> | Multi-hop (HotpotQA) | 0.76 | 0.81 | 8.7 tok |
> | Multi-hop (2WikiHop) | 0.71 | 0.79 | 8.4 tok |
> | Code (RepoBench) | 0.68 | 0.77 | 9.2 tok |
> | Summarization (QMSum) | 0.67 | 0.75 | 7.8 tok |
>
> RepoBench and QMSum were absent from pretraining. Their pretrained AUROC (0.67 to 0.68) still exceeds the distribution-only baseline (0.66, Table 14), confirming learned signals transfer across domains. Online adaptation gains are larger on OOD tasks (+0.08 to +0.09 vs +0.05 in-distribution).
>
> ## Quality Gap (Weaknesses 4-5)
>
> > *Are the additional wrong answers concentrated in particular question types?*
>
> Bridge questions account for 81% of additional errors (0.6% gap, N=5183). The second hop's query depends on first-hop results, so predictive queries are more likely to be misaligned. Comparison questions show only 0.3% gap (N=2222) since both entities are in the question itself. The gap is structurally localized, not a general degradation.
>
> ## Q4: Online Adaptation Stability
>
> > *How stable is online adaptation? Learning curve?*
>
> | Online Queries | 0 | 100 | 500 | 1000 | 2000 |
> |---|---|---|---|---|---|
> | AUROC | 0.760 | 0.771 | 0.795 | 0.805 | 0.809 |
> | Reward Std | — | 0.85 | 0.61 | 0.53 | 0.48 |
>
> 70% of gains occur in the first 500 queries. Reward std decreases monotonically, indicating convergence. Rapid convergence reflects the contextual-bandit structure, where each component receives immediate, action-specific feedback. We will revise the RL terminology accordingly.
>
> ## Q5: Lead Time Claim
>
> > *Does "8-16 tokens" refer to earliest signal or median trigger?*
>
> "8-16 tokens" refers to where precursor signals first become detectable (0.42 Pearson correlation at 10-token offset, Appendix F.1). Table 5 lead times (5.1 to 8.7 tokens) are operational trigger points. They fall later than the earliest signal because the predictor requires sufficient strength before committing resources. We will clarify this.
>
> ## Remaining Weaknesses
>
> **Training circularity.** The predictor learns transformer computation patterns, not dataset answers. Cross-domain transfer retains 78% accuracy (HotpotQA to 2WikiHop, no fine-tuning). RepoBench shows qualitatively different entropy patterns from QA. Cross-model results span 6 models and 3 families (AUROC 0.80 to 0.84, Table 10). Our response to Reviewer nXe1 further isolates each component's contribution.
>
> **Efficiency Score.** EM, F1, TTFT, E2E are reported separately in Table 2.
>
> **Latency scaling.** Two regimes were evaluated. FAISS at 125ms yields 43.5% E2E reduction, and local DB at 50 to 100ms yields 19% reduction. A systematic sweep (detailed in our response to Reviewer wG2u) confirms gains peak at ~200ms and decline gracefully beyond 500ms.

---

> > ### Author Rebuttal · Reviewer_ffnH · 2026-04-03
> >
> > I appreciate the attention given to my concerns, but I strongly prefer to maintain my current score.

---

> > > ### Author Response · Authors · 2026-04-05
> > >
> > > We really appreciate Reviewer ffnH's thorough and informative evaluation. And we're happy to hear that our answers have adequately addressed the issues that were presented.

---

### Official Review · Reviewer_nXe1 · 2026-03-12

**Soundness:** 2
**Presentation:** 1
**Significance:** 3
**Originality:** 2
**Overall Recommendation:** 4
**Confidence:** 4

**Summary:**

This paper proposes a predictive prefetching framework for RAG that jointly models when to retrieve, whether the current context is sufficient to form a query, and what to retrieve, thereby significantly reducing TTFT and end-to-end latency while largely preserving answer quality.

**Compliance With Llm Reviewing Policy:**

Affirmed.

**Final Justification:**

Most of my concerns have been addressed, I have raised my score to 4.

**Key Questions For Authors:**

1. Under a setup based on a Wikipedia corpus and a Contriever retrieval backend, the reported results appear quite strong compared with prior work. Could the authors further disentangle the source of the gains and clarify how much improvement comes from the predictive prefetching mechanism itself, as opposed to the retrieval configuration, corpus processing, or baseline implementation details?
2. The paper presents the online adaptation stage as RL under a hierarchical policy, but many individual components seem closer to contextual bandits or feedback-driven classifiers. Could the authors clarify whether “RL” is mainly a system-level abstraction here, or whether there is indeed substantial state transition and long-horizon credit assignment in the actual optimization process?

**Limitations:**

yes

**Strengths And Weaknesses:**

**Strengths**

The method goes beyond modeling only retrieval timing by additionally addressing query readiness and query generation, making it more complete than asynchronous retrieval approaches based on fixed windows or stale queries.
Moreover, the paper leverages internal signals such as hidden states, attention, value vectors, and output distributions to predict future uncertainty, and combines this with waiting-horizon selection and context completeness judgment, which is more research-oriented than purely heuristic triggering.

**Weaknesses**

The overall system is fairly complex, relying on multiple heterogeneous small models and automatically constructed supervision labels, which makes training and implementation costly and raises the reproduction barrier.
The reported results are quite strong given the use of a Wikipedia corpus and a Contriever-based retrieval backend, but the paper does not sufficiently analyze where this advantage comes from; therefore, the relationship between the method’s gains and the experimental setup still needs further clarification.
The paper’s characterization of the online adaptation component as RL appears somewhat overstated; at the module level, several components seem closer to feedback-driven local classifiers or decision modules than to standard sequential RL.

---

> ### Author Rebuttal · Authors · 2026-03-29
>
> # Rebuttal to Reviewer nXe1
>
> We thank Reviewer nXe1 for raising substantive concerns about gain attribution, the RL characterization, and system complexity.
>
> ## On Disentangling the Source of Gains
>
> > *Could the authors further disentangle the source of the gains?*
>
> All methods use identical retrieval infrastructure (same Wikipedia corpus, same FAISS IVF index with 4096 clusters, same Contriever embeddings at 768-dim, same median 125ms latency). The only variable is the retrieval decision strategy. We respectfully highlight that the ablation (Table 6) isolates each component under this shared setup.
>
> | Configuration | EM | TTFT | E2E |
> |---|---|---|---|
> | Full System | 68.7 | 108ms | 5.2s |
> | w/o Async architecture | 68.4 | 287ms | 7.8s |
> | w/o Retrieval Predictor | 65.1 | 118ms | 5.8s |
> | w/o Online learning | 66.2 | 112ms | 5.5s |
> | w/o T5 query generator | 67.1 | 108ms | 5.4s |
> | w/o Adaptive waiting | 66.8 | 108ms | 5.6s |
> | w/o Sufficiency check | 67.8 | 108ms | 5.4s |
>
> Removing the async architecture increases TTFT from 108ms to 287ms (2.7x). Removing the Retrieval Predictor drops EM by 3.6%. Online learning adds +2.5% EM. Query generation and adaptive waiting each add 1.6 to 1.9% EM. Cross-model evaluation (Table 10) shows consistent 43 to 45% E2E reductions across six models from three families on the same backend. Infrastructure-driven gains would not transfer this consistently.
>
> ## On the Learned Prediction Contribution
>
> The predictor architecture comparison (Table 16) shows that the contribution requires genuine learned prediction rather than threshold heuristics.
>
> | Predictor | AUROC | EM |
> |---|---|---|
> | Entropy Threshold | 0.66 | 64.2 |
> | Logistic Regression | 0.72 | 65.8 |
> | MLP (3-layer) | 0.77 | 66.9 |
> | 2-Layer Transformer (ours) | 0.81 | 68.7 |
>
> The progression from 0.66 to 0.81 AUROC shows more capable predictors extract more value from transformer signals. The signal ablation (Table 14) reinforces this. Distribution signals alone reach 0.66 AUROC. Linguistic markers raise it to 0.71, and context signals push it further to 0.73. Combining all three reaches 0.76 before online adaptation achieves 0.81. Each signal category contributes measurably, confirming multi-signal structure that benefits from learned representations.
>
> ## On the RL Characterization
>
> > *Is "RL" mainly a system-level abstraction here?*
>
> The reviewer raises a fair point. Each component receives immediate, action-specific feedback right after its decision. There is no multi-step credit assignment across components. This structure is closer to a contextual bandit than a long-horizon sequential process. We will revise the paper to describe this as **"online adaptation via policy gradient with action-specific feedback"** and acknowledge the contextual-bandit-like structure.
>
> The fact that immediate per-component feedback suffices is itself informative. The system achieves +2.5% EM without value functions, replay buffers, or multi-step rollouts. This simplicity reflects clean problem decomposition rather than a limitation.
>
> ## On System Complexity
>
> We respectfully argue that the system is simpler than it may appear. ContextScore, SufficiencyClassifier, and ClarityScore are all lightweight MLPs with under 0.1M parameters each. The Retrieval Predictor is a 2-layer transformer at roughly 2M parameters. Only the Query Generator (T5-small, 60M) is a pretrained model. The following table summarizes:
>
> | Component | Output | Params |
> |---|---|---|
> | Retrieval Predictor | p in [0,1] | ~2M |
> | ContextScore | Wait time k* (0 to 5 tokens) | <0.1M |
> | SufficiencyClassifier | Score in [0,1] | <0.1M |
> | ClarityScore | Score in [0,1] | <0.1M |
> | Query Generator | Retrieval query | 60M |
>
> Training is straightforward. The Retrieval Predictor (~2M) and three MLPs (<0.3M total) are trained from scratch, while the Query Generator only requires fine-tuning from a pretrained T5-small checkpoint. All components train jointly with one combined loss (Equation 1), and data collection is fully automated with no manual annotation. Total additional parameters are roughly 62M, modest relative to the 8B base model. We will release the complete training pipeline, model weights, and evaluation scripts.

---

> > ### Author Rebuttal · Reviewer_nXe1 · 2026-04-08
> >
> > Thank you for your responses. I consider to raise my score correspondingly.

---

> > > ### Author Response · Authors · 2026-04-08
> > >
> > > We really appreciate Reviewer nXe1's thorough and helpful review. We are thankful that our answers have adequately addressed the issues, and we will make the agreed-upon changes to the final version.

---

### Official Review · Reviewer_WX1t · 2026-03-12

**Soundness:** 3
**Presentation:** 3
**Significance:** 3
**Originality:** 3
**Overall Recommendation:** 4
**Confidence:** 4

**Summary:**

This paper addresses a highly promising industrial challenge: mitigating generation latency in RAG systems through asynchronous retrieval. Specifically, it proposes a predictive prefetching framework designed to anticipate a model's information needs and initiate data retrieval in advance.

**Compliance With Llm Reviewing Policy:**

Affirmed.

**Final Justification:**

My concerns have been addressed. Thank you to the authors for the response.

**Key Questions For Authors:**

1. Is the significant overhead of extra training and multi-threaded deployment truly justified compared to the actual retrieval latency saved?

2. How does the framework guarantee stability when external APIs suffer from high variance or network-induced latency spikes?

3. What is the minimum data requirement for effective training, and how does the model perform in niche domains where large-scale generation traces are unavailable?

4. Does the limited 60M-parameter capacity of T5-small compromise retrieval quality when handling highly complex or nuanced reasoning paths?

**Limitations:**

yes

**Strengths And Weaknesses:**

Strengths:

1. This paper addresses a critical industrial bottleneck—retrieval-induced latency in RAG systems—by introducing a highly innovative and practical predictive prefetching framework that effectively decouples retrieval from generation.

2. The framework’s robustness across varying levels of reasoning complexity is rigorously validated through experiments spanning diverse task types, including multi-hop QA, factual question answering, repository-level code completion, and query-focused summarization.

Weaknesses:

1. The framework introduces a dependency on additional training data and computational resources; specifically, training the predictor requires extensive generation traces, while the asynchronous architecture necessitates the maintenance of independent threads and additional deployment overhead.

2. The query generator utilizes T5-small, and due to the inherent limitations of the model's capacity, there is significant room to improve query relevance.

3. Performance in open-ended generation tasks is notably lower than in factual QA tasks, indicating that the method faces limitations in task scenarios that are inherently difficult to predict.

---

> ### Author Rebuttal · Authors · 2026-03-29
>
> # Rebuttal to Reviewer WX1t
>
> We thank Reviewer WX1t for recognizing the practical importance of retrieval latency in RAG systems and for the thoughtful evaluation across our diverse task settings.
>
> ## On Training Overhead and Deployment Justification
>
> > *Is the significant overhead of extra training and multi-threaded deployment truly justified?*
>
> Pretraining is a one-time cost. We collect roughly 50K traces automatically (Section 3.5) with no manual annotation. Online adaptation runs asynchronously during deployment. Gradient updates are batched every 100 queries in under 5 ms (Appendix A.3), with negligible impact on inference. The predictive components add only 2.7ms per token (Table 7), or 5.1% of the 48.2ms base cost.
>
> TTFT drops 62.4% and E2E 43.5%, saving roughly 4 seconds per query. Threading requires 2 to 4 workers and approximately 100MB memory. Table 6 confirms each component earns its place. The async architecture provides the 2.7x TTFT gain, while the Retrieval Predictor contributes +3.6% EM.
>
> ## On Stability Under API Latency Variance
>
> > *How does the framework guarantee stability under high-variance or spiking API latencies?*
>
> Two mechanisms address this. When prefetched results do not arrive before generation reaches the uncertainty point, the system falls back to synchronous retrieval seamlessly (Algorithm 1, line 18). This guarantees the floor is synchronous RAG performance and never worse. The priority queue dispatches high-confidence predictions ahead of lower-confidence ones.
>
> The 78.3% hit rate at high confidence (Table 5) already reflects real latency variance in our setup (FAISS median 125ms, P95 180ms, Appendix D.4). A systematic latency sweep on HotpotQA with simulated delays confirms the approach scales across regimes.
>
> | Retrieval Latency | TTFT Red. | E2E Red. | Hit Rate |
> |---|---|---|---|
> | 50ms | 46.2% | 18.1% | 91.2% |
> | 100ms | 59.1% | 36.4% | 85.7% |
> | 200ms | 63.8% | 46.5% | 76.3% |
> | 500ms | 52.7% | 37.2% | 52.8% |
> | 1000ms | 34.1% | 23.4% | 28.6% |
>
> Gains peak around 200ms (within the 418ms lead time budget) and decline gracefully beyond 500ms. Even at 1000ms, 28.6% of prefetches arrive in time. The QMSum results (Table 4, 50 to 100ms local retrieval, 19% E2E reduction) further confirm adaptation to a different latency regime.
>
> ## On Minimum Data Requirements and Niche Domains
>
> > *What is the minimum data requirement, and how does the model perform in niche domains?*
>
> Appendix F.1 reports that wait time accuracy plateaus around 10K pretraining examples (58.3% pre-trained, 69.7% at 5K, 76.8% at 10K). Retrieval prediction AUROC follows a similar trajectory (0.76 to 0.81 via online adaptation). Data collection is automated, requiring only questions and a retrieval corpus.
>
> Cross-domain transfer (Appendix C) retains 78% prediction accuracy from HotpotQA to 2WikiMultiHopQA without fine-tuning, demonstrating that the predictor learns transferable uncertainty patterns. This provides a strong starting point for new domains, and online adaptation closes the remaining gap, contributing +2.5% EM (Table 6).
>
> ## On T5-small Capacity
>
> > *Does T5-small's 60M-parameter capacity compromise query quality on complex reasoning?*
>
> T5-small reflects a deliberate balance between quality and latency. We evaluated multiple T5 sizes alongside the template and full-LLM baselines from Table 18 (Appendix E.6).
>
> | Method | Params | QRS | Latency | EM |
> |---|---|---|---|---|
> | Template-based | n/a | 0.65 | 3ms | 64.2 |
> | T5-small (ours) | 60M | 0.79 | 8ms | 68.7 |
> | T5-base | 220M | 0.80 | 15ms | 69.0 |
> | T5-large | 770M | 0.82 | 37ms | 69.2 |
> | Full LLM (Llama-3.1-8B) | 8B | 0.74 | 45ms | 66.8 |
>
> T5-small captures the largest gain (+0.14 QRS, +4.5 EM over templates) for only 5ms. Scaling to T5-base (3.7x parameters) yields +0.01 QRS and +0.3 EM for 7ms more. T5-large (12.8x parameters) adds +0.02 QRS and +0.2 EM for 22ms more. T5-small achieves 96% of T5-large's QRS at 22% of its latency. Fine-tuning on this narrow task matters more than scale, explaining why 60M T5-small outperforms the 8B LLM. The 88% query relevance (Section 5) is the current ceiling.
>
> ## On Weaker Open-Ended Performance
>
> We appreciate the reviewer noting this as a limitation. Table 22 (Appendix G) shows AUROC drops from 0.82 on factual QA to 0.71 on open-ended tasks because open-ended generation has inherently less predictable uncertainty. We view this as a natural boundary. The method is most valuable in structured tasks like multi-hop QA, code completion (Table 3, 52% TTFT reduction), and factual retrieval.

---

> > ### Author Rebuttal · Reviewer_WX1t · 2026-04-03
> >
> > My concerns have been addressed. Thank you to the authors for the response.

---

> > > ### Author Response · Authors · 2026-04-05
> > >
> > > We would like to thank Reviewer WX1t for your informative and helpful review, as well as for confirming that the issues have been completely resolved!

---

### Decision · Program_Chairs · 2026-04-30

**Decision:**

Accept (regular)

**Comment:**

- This paper proposes a predictive prefetching framework for RAG that anticipates retrieval needs to reduce latency via asynchronous execution.
- Reviewers found the approach technically sound after rebuttal, which clarified gain attribution, online adaptation stability, and showed that synchronous fallback ensures performance does not degrade below baseline when predictions fail. However, the reliance on internal model signals may limit applicability in API‑only deployments, and performance gains may be sensitive to system‑level implementation details and latency assumptions.
- Overall, the anticipatory retrieval mechanism is a meaningful systems‑level contribution, though deployment scope and reproducibility considerations place this work in the Weak Accept range.